# QUANTY, a quantum many-body scripting toolkit

Kevin Ackermann[1], Kai Arnold[1,2], Martin Braß[1,3], Charles Cardot[4],
Robert Green[5,6], Simon Heinze[1], Paul Hill[7], Yi Lu[1,8,9], Sebastian Macke[10,11],
Marius Retegan[12], Sina Shokri[1], Michelangelo Tagliavini[1], Aleksandrs Zacinskis[1], and
Maurits W. Haverkort [1*]

**1** Institute for Theoretical Physics, Heidelberg University, Philosophenweg 19, 69120
Heidelberg, Germany
**2** Department of Mechanical Engineering, The University of Melbourne, Parkville, VIC 3010,
Australia
**3** Institute of Solid State Physics, TU Wien, 1040 Vienna, Austria
**4** Physics Department, University of Washington, Seattle, WA 98105, United States
**5** Department of Physics & Engineering Physics, University of Saskatchewan, Saskatoon,
Saskatchewan, Canada S7N 5E2
**6** Stewart Blusson Quantum Matter Institute, University of British Columbia, Vancouver,
British Columbia, Canada V6T 1Z1
**7** Physikalisches Institut der Universität Heidelberg, Im Neuenheimer Feld 226, 69120
Heidelberg, Germany
**8** National Laboratory of Solid State Microstructures and Department of Physics, Nanjing
University, Nanjing 210093, China
**9** Collaborative Innovation Center of Advanced Microstructures, Nanjing University, Nanjing
210093, China
**10** Max Planck Institute for Solid State Research, Heisenbergstraße 1, 70569, Stuttgart,
Germany
**11** Quantum Matter Institute and Department of Physics and Astronomy, University of British
Columbia, 2355 East Mall, Vancouver, V6T 1Z4, Canada
**12** European Synchrotron Radiation Facility (ESRF), CS40220, 38043, Grenoble Cedex,
France
\* M.W.Haverkort@thphys.uni-heidelberg.de

December 23, 2024

## Abstract

**We introduce version 0.8 of QUANTY, a Lua-based scripting language for many-body physics, written in C/C++. QUANTY enables users to script many-body Hamiltonians for fermionic quantum problems using second quantization. Eigenstates and (non-linear) response functions of these Hamiltonians can be computed when feasible. The software can import tight-binding outputs from various Density Functional Theory or Hartree-Fock codes, allowing the generation of *ab initio* material-specific models. These models can then be used as a starting point for calculations fitted to experimental data. QUANTY implements a range of numerical methods that can be used without requiring a deep understanding of the numerical intricacies of the implemented functions.**

# 1   Introduction

QUANTY provides a versatile scripting language based on Lua 5.2 [1,2]. QUANTY has been used to advance several spectroscopic techniques and in general allows the user to solve problems in many-body quantum physics. The code was originally developed to calculate spectroscopy on correlated open $d$- and $f$-shell materials. Early applications looked at nonresonant inelastic x-ray scattering [3–6], optical Mott-Hubbard and charge transfer excitations [7,8], or resonant inelastic x-ray scattering [9–11]. Over time, the range of problems for which the code is applied to has grown and now also includes calculations of, non-linear pump-probe spectroscopy [12,13], electron capture nuclear decay spectroscopy [14,15], high-precision atomic electron binding energies [16], magnetic susceptibility [17], and a wide range of different types of spectroscopy including, x-ray absorption spectroscopy [18–21], resonant x-ray diffraction [22–24], x-ray emission spectroscopy [25], core-level photoelectron spectroscopy [26], and valence band photoelectron spectroscopy [27,28]. Where earlier publications discussing the code only focused on specific numerical aspects of the algorithms implemented [29,30], this paper describes the basic philosophy of the code and its general underlying structure.

# 2   Setup and philosophy

## 2.1   Lua based scripting

The main part of QUANTY implements several routines to calculate static and dynamic properties of quantum many-body systems. These routines are made accessible by including them as functions in Lua. Lua is a lightweight scripting language developed at the Pontifical Catholic University of Rio de Janeiro in Brazil [1, 2]. In Lua you can perform advanced number arithmetic, evaluate boolean logic, make loops, and use, if-then-else, constructs. All functions and external libraries available in and for Lua can be used in QUANTY. In addition to this, QUANTY adds objects such as `Wavefunctions`, `Operators`, and `ResponseFunctions` to the language. These objects represent quantum states, quantum operators, and measurements. QUANTY furthermore implements functions that allow one to find eigenstates of operators, calculate expectation values of operators and wavefunctions, or calculate response functions.

A single function in the interface can have several underlying numerical implementations. Depending on the problem at hand one implementation might be more efficient than another. For example, the function `Eigensystem` calculates the lowest $N$ eigenstates of an operator. Internally, it can represent an operator as a dense matrix and call a lapack library to diagonalize the matrix. The function `Eigensystem` can also use an internal sparse representation of operators and wave functions and use a shifted re-starting Lanczos routine to calculate the lowest few eigenstates. The actual underlying routine used depends on the problem at hand. In the case of the function `Eigensystem`, the size of the Hilbert space of the physics problem determines whether `Eigensystem` will choose a dense or sparse algorithm. Each function accepts options such that the user has full control over the internal algorithm used. The general idea is that users can use QUANTY without the need to fully understand the underlying algorithms and can focus on using QUANTY to solve problems in quantum physics and chemistry.

## 2.2   Historical background and recent scientific output

The core of the code has been developed in the early 2000's in Mathematica to study basic quantum mechanics and atomic spectroscopy. The Mathematica version of QUANTY is still available [31] and is useful for visualizing charge densities and results of smaller problems. The first scientific publications focused on spectroscopy on correlated materials [3,7,32,33].

In particular, studies focused on the types of excitations that can be understood with relatively local atomic-like physics. Theoretical models used in QUANTY are often crystal-field theory, or ligand-field theory [34]. In the former case, the solid is approximated by a single correlated open-shell atom in an effective potential. In the latter case, the solid is approximated by a single correlated open-shell atom interacting with a bath. The bath states in ligand field theory are often called ligand orbitals. The model is made tractable by including only two particle interactions on the open-shell atom, by reducing the number of ligand sites to very low numbers (1 or 2), and by restricting the number of excitations into the ligand orbitals. Several other codes are available that can perform full multiplet ligand field theory or full multiplet dynamical mean field theory calculations for spectroscopy [35–45]. Ligand field theory, as for example described in the textbook of Ballhausen [34] is also called charge-transfer multiplet ligand field theory or cluster calculations. Most notably the code developed by Cowan, Butler, and Thole [36–38] and maintained by Frank M. F. de Groot [46] and the XTLS code developed by Arata Tanaka [39] were available at the time the development of QUANTY started. Some reasons for developing QUANTY from a tool where we tested how theories work to a full program was to be able to handle systems with low crystal symmetry [10] with parameters based on density functional theory [29] and to go beyond the discretization of a bath at a few bath sites (typically 1 or 2) to several hundred discretization sites [30, 47, 48]. This was achieved by going to natural impurity orbitals [30, 47, 48] and pruning states by removing determinants with a small contributing weight [29]. Version 0.8 makes these algorithms widely available in the Lua based interface. Version 0.8 implements several new functions and new structures for response functions that can easily handle the needed transformations, as described in the main body of this paper.

The implemented methods are used mainly for correlated materials with open $d$- or $f$-shells. These materials show an interaction between the atomic multiplet states and band-formation or charge transfer covalence. This includes $3d$, $4d$, and $5d$ transition metal compounds that can be used, for example, in batteries, solar cells, or catalysis. It also includes compounds with open $4f$- or $5f$-shells that are used in magnetism and show heavy fermion and Kondo physics. More specifically QUANTY has recently been used to describe a variety of experiments including time dependent pump-probe spectroscopy [13, 49], x-ray magnetic circular dichroism and other polarization dependent core level x-ray absorption spectroscopy [50–68], non-resonant inelastic x-ray scattering [66], core level photo-electron spectroscopy [52, 66, 69, 70], core to core and core to valence resonant elastic and inelastic x-ray scattering [52,56,71–75] including dispersion of low energy magnetic excitations, resonant energy dependence and polarization dependence, x-ray emission spectroscopy [76,77], valence band photo electron spectroscopy [78–80], including polarization dependent band intensities, electron capture nuclear decay spectra [81], and high precision mass spectroscopy [16]. These experiments are performed on different classes of materials including metallic lanthanide compounds [51, 66, 69] showing possible heavy fermion and Kondo physics, molecular coordination complexes and nano particles [50, 57, 58, 60, 70, 73] with possible persistent single-ion magnetic ordering, strong magnetic anisotropy, or catalytic properties, transition metal centers in Li containing battery materials [52, 53, 59, 72, 77], transition metal metal compounds [54, 56, 61–65, 67, 68, 75, 78–80], including high entropy metals, magnetic materials, 2D transition metal dichalcogenides, battery and solar cell materials, actinide materials [74], and highly charged well isolated atomic ions [16] that can be used to test the fundamental microsocpic theories of physics as formulated within the standard model. In addition to material-specific calculations, QUANTY has been used to study different models [82] that can be used to understand possible physics in correlated materials.

## 2.3 Compilation and licenses

QUANTY is written in C and C++. Statically linked binaries for Windows, Linux, and MacOS, can be downloaded from www.quanty.org [31]. The source code is available at git.quanty.org. In order to compile QUANTY with the Lua interface, one needs to compile a single file. This file, *main.cpp*, can be found in the folder *Luainterface*. The code needs to be linked to external linear algebra libraries, BLAS and LAPACK. Furthermore, one needs to specify if one compiles for a Linux, Mac, or Windows environment by setting the appropriate compile-time variable. The use of a single file to compile combined with the use of a minimal number of external libraries allows the code to be easily compiled on a wide variety of different machines and setups independent of the installed operating system, the installed C++ compiler, or installed packages.

The source code includes a slightly modified copy of Lua. The Lua-derived code in the folder *Luainterface/lua* is licensed under the MIT license. In addition to the embedded Lua source code, a small subset of functions in the QUANTY codebase are copied from other developers. For example, QUANTY implements several random number generators that can be used to test their randomness and efficiency against each other. Copyright notices for these routines [83] are included before implementing the actual source code. The main part of QUANTY is released under a CC-BY 4.0 license. Users can freely use and adapt QUANTY and its embedded 3rd party software for scientific and commercial use under the CC-BY 4.0 license.

# 3 Physics, Chemistry, and numerical background

The following section describes, in a very compact fashion, the basic underlying mathematical structure of quantum many-body calculations as used in QUANTY. In addition, it introduces the notation that we will use in later sections of the paper.

## 3.1 Quantum states

The microstate of an $n$ particle system is, in classical physics, given by the precise position and momenta of all particles. In quantum mechanics, the state is described by an $n$ particle wave function that assigns a complex number (vector) to all possible values of the $3n$ coordinates of all particles. We generally do not work with wave functions in quantum mechanics directly, but we use the fact that states in quantum mechanics can be written as sums over a set of basis states. We need to distinguish the one-particle basis states and the $n$-particle basis states.

### 3.1.1 One particle basis states

We start by defining a basis of $N_F$ one-particle fermionic basis states and $N_B$ one-particle bosonic basis states. The current release version requires $N_B = 0$, as bosons are only partially implemented. As such, we will discuss only fermionic particles. We can label the basis of one-particle quantum states as $|\phi_\alpha\rangle$ with $\alpha \in [0, \ldots, N_F - 1]$. The states $|\phi_\alpha\rangle$ can be represented by a function

$$\phi_\alpha(\vec{r}) = \langle r | \phi_\alpha \rangle. \tag{1}$$

For non-relativistic electrons with spin 1/2 these functions assign a complex vector of length 2 to each point in space related to the spin density of the electron. For full relativistic calculations using the Dirac equation, the one-particle wave function assigns a vector with 4 complex components to each point in space representing the Dirac spinors.

A general one-particle state $|\varphi_\tau\rangle$ can be written as a linear combination of the basis states.

$$|\varphi_\tau\rangle = \sum_{\alpha=0}^{N_F-1} c_\tau^{(\alpha)} |\phi_\alpha\rangle, \tag{2}$$

with $c_\tau^{(\alpha)}$ a complex number. For $N_F$ basis states, one can define $N_F$ different, linearly independent, states $|\varphi_\tau\rangle$.

### 3.1.2  Second quantization

We can replace states with operators that create or annihilate particles. This has the advantage that the anti-symmetrization of $n$ particle wave functions can be absorbed in the anti-commutation relations of the creation and annihilation operators. We define the state $|0\rangle$ to be the vacuum state, that is, a state with 0 particles. The operator $a_\tau^\dagger$ creates a particle in the one particle state $|\varphi_\tau\rangle$ such that

$$a_\tau^\dagger |0\rangle = |\varphi_\tau\rangle. \tag{3}$$

We can create a two-particle state by successively acting with two creation operators. The wave-function that represents this state depends on two coordinates. The wave-function needs to be antisymmetric under the exchange of the positions of particle 1 and particle 2. We find the equivalent notations:

$$a_{\tau_1}^\dagger a_{\tau_2}^\dagger |0\rangle = \left|\varphi_{\tau_1}\varphi_{\tau_2}\right\rangle \Longleftrightarrow \tag{4}$$

$$\psi(\vec{r}_1, \vec{r}_2) = \sqrt{1/2}\left(\varphi_{\tau_1}(\vec{r}_1)\varphi_{\tau_2}(\vec{r}_2) - \varphi_{\tau_1}(\vec{r}_2)\varphi_{\tau_2}(\vec{r}_1)\right). \tag{5}$$

In order for these two notations in equation 4 and 5 to be equivalent we require the creation operators to anti-commute

$$a_{\tau_1}^\dagger a_{\tau_2}^\dagger = -a_{\tau_2}^\dagger a_{\tau_1}^\dagger. \tag{6}$$

An $n$ particle wave function has $n!$ terms. At the same time, the state it represents can be written as a single product of $n$ creation operators.

In addition to the creation operators $a_\tau^\dagger$ that create a particle in the one-particle state $|\varphi_\tau\rangle$, we also define the annihilation operators $a_\tau$ that annihilate a particle from the state $|\varphi_\tau\rangle$.

### 3.1.3  n-particle single Slater determinant basis states

Given a one-particle Hilbert space with a basis of $N_F$ one particle states $|\varphi_\tau\rangle$, we can define an $n$-particle state $\left|\psi_j\right\rangle$ by selecting $n$ basis states and successively creating particles in these states.

$$\left|\psi_j\right\rangle = \left(\prod_{\tau \in D_j} a_\tau^\dagger\right)|0\rangle, \tag{7}$$

with $D_j$ a set of $n$ one particle states chosen from the $N_F$ possible one particle basis states such that $\tau \in [0, \ldots, N_F - 1]$. There are in total

$$\text{Dim}(\mathcal{H}_{n,N_F}) = \binom{N_F}{n} = \frac{N_F!}{n!(N_F-n)!}, \tag{8}$$

possible sets $D_j$. These are the different ways to choose $n$ different states from the $N_F$ basis states. The resulting $\text{Dim}(\mathcal{H}_{n,N_F})$ different $n$-particle states $\left|\psi_j\right\rangle$ define the $n$-particle Hilbert

space $\mathcal{H}_{n,N_F}$. This number becomes large quickly, which is one of the main bottlenecks in many-body quantum computations.

The states $\left|\psi_j\right\rangle$ are known as single Slater determinant states. Single Slater-determinant states are not the general eigenstates of a quantum many-body problem. On a mean-field level such as within the Hartree-Fock or Kohn-Sham density functional theory approximation one can write all eigenstates as single Slater determinants. For correlated quantum systems, this is not sufficient.

### 3.1.4 General n-particle states

Given $N_F$ one particle basis states and $n$ electrons, one can define $\binom{N_F}{n}$ $n$-particle single Slater determinant states. These states define a basis for a general $n$-particle state. We can write a general $n$-particle state $|\Psi_\alpha\rangle$ as

$$|\Psi_\alpha\rangle = \sum_{j=1}^{\binom{N_F}{n}} c_j^{(\alpha)} \left|\psi_j\right\rangle, \tag{9}$$

with $c_j^{(\alpha)}$ a set of $\binom{N_F}{n}$ complex numbers defining the expansion coefficients of the $n$-particle state on the $n$-particle basis.

## 3.2 Operators

Any operator in quantum mechanics can be written as a sum over products of creation and annihilation operators. The translation between operators in second quantization and first quantization is straight forward. For most problems, we can separate the operators into one- and two-particle operators. One-particle operators act on a single particle coordinate. These are operators such as the kinetic energy operator or an external potential. Two particle operators act on two coordinates at the same time. These operators include the Coulomb interaction or the total angular momentum operators $L^2$ of a system. Operators that act on 3 or more particles simultaneously can be implemented and used in QUANTY. Multi-particle operators, for example, appear in renormalization group techniques as effective interactions, but do not appear in the Schrödinger or Dirac equation.

For one electron operators we find

$$H_1 = \sum_{\tau_1,\tau_2} \epsilon_{\tau_1,\tau_2} a_{\tau_1}^\dagger a_{\tau_2}, \tag{10}$$

with

$$\epsilon_{\tau_1,\tau_2} = \int_{\mathbb{R}^3} \phi_{\tau_1}^*(\vec{r}) H_1(\vec{r}) \phi_{\tau_2}(\vec{r}) \mathrm{d}\vec{r}. \tag{11}$$

For two electron operators we have

$$H_2 = \sum_{\tau_1,\tau_2,\tau_3,\tau_4} U_{\tau_1,\tau_2\tau_3,\tau_4} a_{\tau_1}^\dagger a_{\tau_2}^\dagger a_{\tau_3} a_{\tau_4}, \tag{12}$$

with

$$U_{\tau_1,\tau_2\tau_3,\tau_4} = \frac{1}{2} \int_{\mathbb{R}^3} \int_{\mathbb{R}^3} \phi_{\tau_1}^*(\vec{r}_1) \phi_{\tau_2}^*(\vec{r}_2) H_2(\vec{r}_1,\vec{r}_2) \phi_{\tau_4}(\vec{r}_1) \phi_{\tau_3}(\vec{r}_2) \mathrm{d}\vec{r}_1 \mathrm{d}\vec{r}_2. \tag{13}$$

### 3.3 Response functions

Measurements in physics and chemistry are based on observing how a system reacts by making a small perturbation to the system. The theory that describes this is response theory. For an unperturbed system defined by the Hamiltonian $H_0$ we can define the time-dependent Hamiltonian $H_1(t)$ that describes the perturbation we make. For example, the Hamiltonian $H_1(t)$ can describe the interaction with photons if we look at an optical absorption experiment or the interaction with neutrons if we look at inelastic neutron scattering. If we act with an oscillating field on our system, we can use response theory in the frequency / energy domain. In this case we can write $H_1(t)$ as a time independent operator $T$ times an oscillating field, $H_1 = T \sin(\omega t)$, with angular frequency $\omega$ and particle energy $\hbar \omega$. The resulting complex linear response function is given as

$$\sigma(\omega) \propto \lim_{\Gamma \to 0^+} \left( \left\langle \Psi_0 \left| T^\dagger \frac{1}{\hbar\omega - H_0 + E_0 + \mathrm{i}\frac{\Gamma}{2}} T \right| \Psi_0 \right\rangle - \left\langle \Psi_0 \left| T^\dagger \frac{1}{\hbar\omega + H_0 - E_0 + \mathrm{i}\frac{\Gamma}{2}} T \right| \Psi_0 \right\rangle \right), \tag{14}$$

with $E_0 = \langle \Psi_0 | H | \Psi_0 \rangle$ and $\Gamma$ a small positive number such that all poles of $\sigma(\omega)$ are below the real axis. The dissipation or absorption coefficient $\alpha$ in a system is given by

$$\alpha = -\mathrm{Im}(\sigma). \tag{15}$$

The limit of $\Gamma$ to $0^+$ can only be taken if a suitable set of continuum states and interactions to these states are included in $H_0$. Although it is possible to include such continuum states in QUANTY [81, 84], in practice we often do not include the coupling to photon fields and the possible fluorescence decay, nor the coupling to free-electron states and the possible Auger-Meitner decay. We often assume that $\Gamma$ is a positive constant in our calculations. The finite value of $\Gamma$ then refers to the life-time broadening of the excited states.

Both terms in equation 14 are in general important. For cases where $\omega \gg \Gamma$ we can neglect the second term. Even more so, if we define

$$G_T(\omega, \Gamma) = \left\langle \Psi_0 \left| T^\dagger \frac{1}{\hbar\omega - H_0 + E_0 + \mathrm{i}\frac{\Gamma}{2}} T \right| \Psi_0 \right\rangle, \tag{16}$$

then

$$\sigma(\omega) \propto G_T(\omega, \Gamma) + G_T(-\omega, \Gamma)^*. \tag{17}$$

Thus, if we know $G_T(\omega, \Gamma)$ we can recreate the linear response function $\sigma(\omega)$. As such we calculate $G_T(\omega, \Gamma)$ in QUANTY.

For nonlinear response one needs to look at higher order expansions of the form

$$G_{T_1, T_2}(\omega_1, \Gamma_1, \omega_2, \Gamma_2) = \tag{18}$$
$$\left\langle \Psi_0 \left| T_1^\dagger \frac{1}{\hbar\omega_1 - H_0 + E_0 - \mathrm{i}\frac{\Gamma_1}{2}} T_2^\dagger \frac{1}{\hbar\omega_2 - H_0 + E_0 + \mathrm{i}\frac{\Gamma_2}{2}} T_2 \frac{1}{\hbar\omega_1 - H_0 + E_0 + \mathrm{i}\frac{\Gamma_1}{2}} T_1 \right| \Psi_0 \right\rangle.$$

### 3.3.1 Relation to Fermi's golden rule and Kramers Heisenberg equation

Fermi's golden rule and the Kramers-Heisenberg equation, which describe the absorption and inelastic scattering of light, can be obtained as approximations of the linear and quadratic response functions. These approximations are often sufficient, and one finds Fermi's golden rule and the Kramers-Heisenberg equation often as the starting point for describing spectroscopy.

We can derive Fermi's golden rule from the linear response of a system. If we neglect the second term in equation 14 we find

$$\alpha \propto -\lim_{\Gamma \to 0^+} \text{Im} \left\langle \Psi_0 \left| T^\dagger \frac{1}{\hbar\omega - H_0 + E_0 + i\frac{\Gamma}{2}} T \right| \Psi_0 \right\rangle \tag{19}$$

We now add a complete set of eigenstates of $\Psi_f$ with eigenenergies $E_f$ to get

$$\alpha \propto -\lim_{\Gamma \to 0^+} \text{Im} \left\langle \Psi_0 \left| T^\dagger \sum_f |\Psi_f\rangle\langle\Psi_f| \frac{1}{\hbar\omega - H_0 + E_0 + i\frac{\Gamma}{2}} \sum_f |\Psi_f\rangle\langle\Psi_f| T \right| \Psi_0 \right\rangle \tag{20}$$

$$= -\sum_f \lim_{\Gamma \to 0^+} \text{Im} \langle\Psi_0|T^\dagger|\Psi_f\rangle \left\langle \Psi_f \left| \frac{1}{\hbar\omega - E_f + E_0 + i\frac{\Gamma}{2}} \right| \Psi_f \right\rangle \langle\Psi_f|T|\Psi_0\rangle \tag{21}$$

$$= \sum_f \langle\Psi_0|T^\dagger|\Psi_f\rangle\langle\Psi_f|T|\Psi_0\rangle \pi\delta(\hbar\omega + E_0 - E_f) \tag{22}$$

whereby we used that $\sum_f |\Psi_f\rangle\langle\Psi_f| = 1$, that $H_0|\Psi_f\rangle = E_f|\Psi_f\rangle$ and that $\lim_{\Gamma \to 0^+} \frac{1}{\hbar\omega - E_f + E_0 + i\frac{\Gamma}{2}}$ is a Dirac delta function. The transition rate between two eigenstates $|\Psi_0\rangle$ and $|\Psi_f\rangle$ is thus given by Fermi's golden rule as $\left|\langle\Psi_0|T|\Psi_f\rangle\right|^2$.

For second-order response theory, we can start from $G_{T_1,T_2}(\omega_1, \Gamma_1, \omega_2, \Gamma_2)$ and derive the Kramers-Heisenberg equation. We define the double differential cross section $\frac{d^2\sigma}{d\Omega_o d\hbar\omega_o}$ as the fraction of light scattered in the solid angle $\Omega_o$ around the Poynting vector $\vec{k}_0$ with photon energy $\hbar\omega_o$ and polarization $\epsilon_o$ after exciting the system with light with photon energy $\hbar\omega_i$, Poynting vector $\vec{k}_i$ and polarization $\epsilon_i$.

The derivation of the Kramers-Heisenberg equation from $G_{T_1,T_2}(\omega_1, \Gamma_1, \omega_2, \Gamma_2)$ is similar to the derivation of Fermi's golden rule for linear response. We start from equation 18 and insert complete sets of eigenstates $|\Psi_f\rangle$ and $|\Psi_n\rangle$.

$$\frac{d^2\sigma}{d\Omega_o d\omega_o} = -\frac{\omega_o}{\omega_i} \frac{1}{\pi} \lim_{\Gamma_f \to 0^+} \text{Im}\, G_{T_{\epsilon_i,\vec{k}_i}, T_{\epsilon_o,\vec{k}_o}}(\omega_i, \Gamma_n, \omega_i - \omega_o, \Gamma_f) \tag{23}$$

$$= -\frac{\omega_o}{\omega_i} \frac{1}{\pi} \lim_{\Gamma_f \to 0^+} \text{Im} \sum_{n_1, n_2, f} \left\langle \Psi_0 \left| T^\dagger_{\epsilon_i,\vec{k}_i} \right| \Psi_{n_1} \right\rangle \left\langle \Psi_{n_1} \left| \frac{1}{\hbar\omega_i - E_{n_1} + E_0 + i\frac{\Gamma_n}{2}} \right| \Psi_{n_1} \right\rangle \tag{24}$$

$$\times \left\langle \Psi_{n_1} \left| T^\dagger_{\epsilon_0,\vec{k}_o} \right| \Psi_f \right\rangle \left\langle \Psi_f \left| \frac{1}{\hbar\omega_i - \hbar\omega_o - E_f + E_0 + i\frac{\Gamma_f}{2}} \right| \Psi_f \right\rangle \left\langle \Psi_f \left| T_{\epsilon_0,\vec{k}_o} \right| \Psi_{n_2} \right\rangle$$

$$\times \left\langle \Psi_{n_2} \left| \frac{1}{\hbar\omega_i - E_{n_2} + E_0 + i\frac{\Gamma_n}{2}} \right| \Psi_{n_2} \right\rangle \left\langle \Psi_{n_2} \left| T_{\epsilon_i,\vec{k}_i} \right| \Psi_0 \right\rangle$$

$$= \frac{\omega_o}{\omega_i} \sum_{n_1, n_2, f} \left\langle \Psi_0 \left| T^\dagger_{\epsilon_i,\vec{k}_i} \right| \Psi_{n_1} \right\rangle \frac{1}{\hbar\omega_i - E_{n_1} + E_0 + i\frac{\Gamma_n}{2}} \left\langle \Psi_{n_1} \left| T^\dagger_{\epsilon_0,\vec{k}_o} \right| \Psi_f \right\rangle \tag{25}$$

$$\times \left\langle \Psi_f \left| T_{\epsilon_0,\vec{k}_o} \right| \Psi_{n_2} \right\rangle \frac{1}{\hbar\omega_i - E_{n_2} + E_0 + i\frac{\Gamma_n}{2}} \left\langle \Psi_{n_2} \left| T_{\epsilon_i,\vec{k}_i} \right| \Psi_0 \right\rangle \delta(\hbar\omega_i - \hbar\omega_o - E_f + E_0)$$

$$= \frac{\omega_o}{\omega_i} \sum_f \left| \sum_n \frac{\left\langle \Psi_f \left| T_{\epsilon_0,\vec{k}_o} \right| \Psi_n \right\rangle \left\langle \Psi_n \left| T_{\epsilon_i,\vec{k}_i} \right| \Psi_0 \right\rangle}{\hbar\omega_i - E_n + E_0 + i\frac{\Gamma_n}{2}} \right|^2 \delta(\hbar\omega_i - \hbar\omega_o - E_f + E_0). \tag{26}$$

Note that polarization and wave-vector dependence enter only via the operator $T$ that couples the initial $|\Psi_0\rangle$, intermediate $|\Psi_n\rangle$ and final states $|\Psi_f\rangle$. It directly follows from the derivation that the sum over all intermediate states is inside the square.

### 3.3.2 Efficient calculation of response functions using Krylov basis sets

In order to calculate the response functions using Fermi's golden rule or the Kramers Heisenberg equation, one needs to know the compete set of all eigenstates of the Hamiltonian $H_0$. This is often a formidable computational task that is not feasible for many interesting systems. Using response functions, one can bypass the need to calculate all the eigenstates of $H_0$ by generating a Krylov basis. The Krylov basis consists of a subset of all states in the many-body basis of $H_0$ on which the spectral function $G_T(\omega, \Gamma)$ can be represented. One can generate the Krylov basis of $G_T(\omega, \Gamma)$ starting from state $T\left|\Psi_0\right\rangle$ using a Lanczos procedure. The Krylov basis of dimension $N_{Tri}$ is spanned by the states $H_0^n T\left|\Psi_0\right\rangle$ for $n \in [0, \ldots, N_{Tri} - 1]$. We can define the orthonormal set of Krylov states $\left|\psi_{KR}^{(n)}\right\rangle$ by the following equations:

$$\left|\tilde{\psi}_{KR}^{(0)}\right\rangle = T\left|\Psi_0\right\rangle \tag{27}$$

$$\left|\tilde{\psi}_{KR}^{(n>0)}\right\rangle = H\left|\psi_{KR}^{(n-1)}\right\rangle - a_n\left|\psi_{KR}^{(n-1)}\right\rangle - b_{n-1}\left|\psi_{KR}^{(n-2)}\right\rangle \tag{28}$$

$$b_n = \sqrt{\left\langle\tilde{\psi}_{KR}^{(n)}\middle|\tilde{\psi}_{KR}^{(n)}\right\rangle} \tag{29}$$

$$\left|\psi_{KR}^{(n)}\right\rangle = \frac{1}{b_n}\left|\tilde{\psi}_{KR}^{(n)}\right\rangle, \tag{30}$$

with $\left|\tilde{\psi}_{KR}^{(-1)}\right\rangle = 0$.

The Hamiltonian on a Krylov basis is given by a tridiagonal matrix with the elements $a_n$ ($n \in [1, \ldots, N_{Tri}]$) on the diagonal and $b_n$ ($n \in [1, \ldots, N_{Tri} - 1]$) on the off diagonal.

$$a_n = \left\langle\psi_{KR}^{(n-1)}\middle|H\middle|\psi_{KR}^{(n-1)}\right\rangle \tag{31}$$

$$b_n = \left\langle\psi_{KR}^{(n)}\middle|H\middle|\psi_{KR}^{(n-1)}\right\rangle. \tag{32}$$

The response function $G_T(\omega, \Gamma)$ is then given by a continued fraction

$$G_T(\omega, \Gamma) = \cfrac{b_0^2}{\omega + i\frac{\Gamma}{2} - a_1 - \cfrac{b_1^2}{\omega + i\frac{\Gamma}{2} - a_2 - \cfrac{b_2^2}{\omega + i\frac{\Gamma}{2} - a_3 - \ldots}}}. \tag{33}$$

The calculation is converged when $b_n = 0$. In practice, a finite number of $N_{\text{Tri}}$ steps is calculated and the changes in the spectral function are viewed as $N_{\text{Tri}}$ increases. The larger $\Gamma$ taken in the calculation, the faster the spectrum converges to the final anser.

Instead of starting from a single state $T\left|\Psi_0\right\rangle$, one can start from several states. This is useful, for example, if one looks at the photoelectron emission spectra of a multi-orbital system, polarization-dependent absorption spectroscopy, temperature dependence, or nuclear decay by electron capture from several different core states. We define $T$ as a list of annihilation operators that act on states 0 to $m-1$. The states $a_\tau\left|\Psi_0\right\rangle$ with $\tau \in [0, \ldots, m-1]$ define a vector of $m$ states. The Lanczos algorithm is replaced by a Block Lanczos algorithm. The resulting response function is still given by a continued fraction. Each level now is an m dimensional matrix

$$G_T(\omega, \Gamma) = B_0^* \cfrac{1}{\omega + i\frac{\Gamma}{2} - A_1 - B_1 \cfrac{1}{\omega + i\frac{\Gamma}{2} - A_2 - B_2 \cfrac{1}{\omega + i\frac{\Gamma}{2} - A_3 - \ldots} B_2^\dagger} B_1^\dagger} B_0^T, \tag{34}$$

with $A_n$ an m dimensional complex Hermitian matrix and $B_m$ can be chosen to be also Hermitian, except for $B_0$ and unless the matrix is rank-deficient.

### 3.3.3 Operators from response functions

For certain approximate calculations, like dynamical mean-field theory, it's necessary to create a noninteracting tight-binding Hamiltonian whose response functions matches the response functions of an interacting model. The concept relies on the fact that solving response functions of non-interacting models is simpler than those of interacting models. An algorithm introduced in [30] generates such a non-interacting Hamiltonian by assuming knowledge of the interacting system's response function. If the response function of an interacting system is obtained through a Lanczos routine, then the response function takes the form of a continuous fraction as specified in equation 33. A one-dimensional tight-binding Hamiltonian for a non-interacting system with the same response function is given as

$$H_T = \sum_{i=1}^{N_{Tri}} a_i \left( a_i^\dagger a_i \right) + \sum_{i=1}^{N_{Tri}-1} b_i \left( a_i^\dagger a_{i+1} + a_{i+1}^\dagger a_i \right), \tag{35}$$

with $b_0 = 1$. One can verify this result by realizing that the tight-binding Hamiltonian can be represented by the following tri-diagonal matrix

$$H_T = \begin{pmatrix} a_1 & b_1 & 0 & 0 & 0 & \dots \\ b_1 & a_2 & b_2 & 0 & 0 & \dots \\ 0 & b_2 & a_3 & b_3 & 0 & \dots \\ 0 & 0 & b_3 & a_4 & b_4 & \dots \\ 0 & 0 & 0 & b_4 & a_5 & \ddots \\ \vdots & \vdots & \vdots & \vdots & \ddots & \ddots \end{pmatrix}. \tag{36}$$

The inverse of a tri-diagonal matrix is a continued fraction.

Being able to represent a response function of an interacting system by a tight-binding Hamiltonian of a noninteracting system is particularly useful for the one-particle Green's function. The local one particle Green's function for site $\tau$ can be created from the response functions one obtains by either creating or annihilating an electron. We define the Green's function for removing an electron from the system, or the Photo Electron Spectroscopy (PES) Green's function as

$$G_{PES}(\omega, \Gamma) = \left\langle \Psi_0 \left| T_{PES}^\dagger \frac{1}{\omega - H + E_0 + i\Gamma/2} T_{PES} \right| \Psi_0 \right\rangle, \tag{37}$$

and Green's function for adding an electron or the Inverse Photo Electron Spectroscopy (IPES) Green's function as

$$G_{IPES}(\omega, \Gamma) = \left\langle \Psi_0 \left| T_{IPES}^\dagger \frac{1}{\omega - H + E_0 + i\Gamma/2} T_{IPES} \right| \Psi_0 \right\rangle, \tag{38}$$

with $T_{IPES} = a_\tau^\dagger$ and $T_{PES} = a_\tau$. More specifically, these response functions are the photoelectron spectroscopy and inverse photoelectron spectroscopy response functions of an interacting system. The one particle Green's function is given as

$$G(\omega) = \lim_{\Gamma \to 0^+} G_{IPES}(\omega, \Gamma) - G_{PES}(-\omega, \Gamma)^*. \tag{39}$$

Using the previously introduced representations we can create a non-interacting tight-binding Hamiltonian $H_G^{(1)}$ that has the same one particle Green's function $G(\omega)$ as the interacting system used to calculate $G_{IPES}(\omega, \Gamma)$ and $G_{PES}(\omega, \Gamma)$, i.e.

$$G(\omega) = \lim_{\Gamma \to 0^+} \left\langle 0 \left| a_\tau \frac{1}{\omega - H_G^{(1)} + i\Gamma/2} a_\tau^\dagger \right| 0 \right\rangle. \tag{40}$$

With the help of a Lanczos routine we can write

$$G_{PES}(\omega,\Gamma) = \cfrac{b_0^{(v)2}}{\omega + i\frac{\Gamma}{2} - a_1^{(v)} - \cfrac{b_1^{(v)2}}{\omega+i\frac{\Gamma}{2}-a_2^{(v)}-...}}, \quad G_{IPES}(\omega,\Gamma) = \cfrac{b_0^{(c)2}}{\omega + i\frac{\Gamma}{2} - a_1^{(c)} - \cfrac{b_1^{(c)2}}{\omega+i\frac{\Gamma}{2}-a_2^{(c)}-...}} \quad (41)$$

Given the tri-diagonal representation of $G_{IPES}(\omega,\Gamma)$ and $G_{PES}(\omega,\Gamma)$ and that $b_0^{(v)2} + b_0^{(c)2} = 1$ we obtain the following matrix representation of the tight-binding Hamiltonian

$$H_G^{(1)} = T \cdot H_{PES,IPES} \cdot T^T, \quad (42)$$

with

$$H_{PES,IPES} = \begin{pmatrix} a_1^{(v)} & 0 & b_1^{(v)} & 0 & 0 & \dots & 0 & 0 & 0 & \dots \\ 0 & a_1^{(c)} & 0 & 0 & 0 & \dots & b_1^{(c)} & 0 & 0 & \dots \\ b_1^{(v)} & 0 & a_2^{(v)} & b_2^{(v)} & 0 & \dots & 0 & 0 & 0 & \dots \\ 0 & 0 & b_2^{(v)} & a_3^{(v)} & b_3^{(v)} & \dots & 0 & 0 & 0 & \dots \\ 0 & 0 & 0 & b_3^{(v)} & a_4^{(v)} & \ddots & 0 & 0 & 0 & \dots \\ \vdots & \vdots & \vdots & \vdots & \ddots & \ddots & \vdots & \vdots & \vdots & \\ 0 & b_1^{(c)} & 0 & 0 & 0 & \dots & a_2^{(c)} & b_2^{(c)} & 0 & \dots \\ 0 & 0 & 0 & 0 & 0 & \dots & b_2^{(c)} & a_3^{(c)} & b_3^{(c)} & \dots \\ 0 & 0 & 0 & 0 & 0 & \dots & 0 & b_3^{(c)} & a_4^{(c)} & \ddots \\ \vdots & \vdots & \vdots & \vdots & \vdots & & \vdots & \vdots & \ddots & \ddots \end{pmatrix}, \quad (43)$$

and

$$T = \begin{pmatrix} b_0^{(v)} & b_0^{(c)} & 0 & 0 & 0 & \dots & 0 & 0 & 0 & \dots \\ -b_0^{(c)} & b_0^{(v)} & 0 & 0 & 0 & \dots & 0 & 0 & 0 & \dots \\ 0 & 0 & 1 & 0 & 0 & \dots & 0 & 0 & 0 & \dots \\ 0 & 0 & 0 & 1 & 0 & \dots & 0 & 0 & 0 & \dots \\ 0 & 0 & 0 & 0 & 1 & \ddots & 0 & 0 & 0 & \dots \\ \vdots & \vdots & \vdots & \vdots & \ddots & \ddots & \vdots & \vdots & \vdots & \\ 0 & 0 & 0 & 0 & 0 & \dots & 1 & 0 & 0 & \dots \\ 0 & 0 & 0 & 0 & 0 & \dots & 0 & 1 & 0 & \dots \\ 0 & 0 & 0 & 0 & 0 & \dots & 0 & 0 & 1 & \ddots \\ \vdots & \vdots & \vdots & \vdots & \vdots & & \vdots & \vdots & \ddots & \ddots \end{pmatrix}. \quad (44)$$

In operator form this yields the Hamiltonian

$$
\begin{aligned}
H_G^{(1)} = {} & \left( a_1^{(c)} b_0^{(c)2} + a_1^{(v)} b_0^{(v)2} \right) a_0^\dagger a_0 + \left( a_1^{(c)} - a_1^{(v)} \right) b_0^{(c)} b_0^{(v)} a_0^\dagger a_1 \\
& + \left( a_1^{(c)} - a_1^{(v)} \right) b_0^{(c)} b_0^{(v)} a_1^\dagger a_0 + \left( a_1^{(c)} b_0^{(v)2} + a_1^{(v)} b_0^{(c)2} \right) a_1^\dagger a_1 \\
& + b_0^{(v)} b_1^{(v)} \left( a_0^\dagger a_2 + a_2^\dagger a_0 \right) - b_0^{(c)} b_1^{(v)} \left( a_1^\dagger a_2 + a_2^\dagger a_1 \right) \\
& + b_0^{(c)} b_1^{(c)} \left( a_0^\dagger a_{N_{Tri}^{(v)}+1} + a_{N_{Tri}^{(v)}+1}^\dagger a_0 \right) + b_0^{(v)} b_1^{(c)} \left( a_1^\dagger a_{N_{Tri}^{(v)}+1} + a_{N_{Tri}^{(v)}+1}^\dagger a_1 \right) \\
& + \sum_{i=2}^{N_{Tri}^{(v)}} a_i^{(v)} \left( a_i^\dagger a_i \right) + \sum_{i=2}^{N_{Tri}^{(v)}-1} b_i^{(v)} \left( a_i^\dagger a_{i+1} + a_{i+1}^\dagger a_i \right) \\
& + \sum_{i=2}^{N_{Tri}^{(c)}} a_i^{(c)} \left( a_{i+N_{Tri}^{(v)}-1}^\dagger a_{i+N_{Tri}^{(v)}-1} \right) + \sum_{i=2}^{N_{Tri}^{(c)}-1} b_i^{(c)} \left( a_{i+N_{Tri}^{(v)}-1}^\dagger a_{i+N_{Tri}^{(v)}} + a_{i+N_{Tri}^{(v)}}^\dagger a_{i+N_{Tri}^{(v)}-1} \right).
\end{aligned}
\tag{45}
$$

The many-electron ground-state of this Hamiltonian is given by the function

$$
|\Psi_0\rangle = \left( b_0^v a_0^\dagger - b_0^c a_1^\dagger \right) \prod_{i=2}^{N_{Tri}^{(v)}} a_i^\dagger |0\rangle .
\tag{46}
$$

Note that for a system with multiple local degrees of freedom, the Lanczos routines are replaced by block Lanczos routines, and the response functions become matrix functions. In this case, each one-particle state with index $i$ is replaced by $m$ states, whereby $m$ is the dimension of the matrix that represents the response or Green's function. The general structure of the operators and matrices remains conserved. In particular, and usefull for the stability of the algorithm, each matrix $A_i$ and $B_i$ can be made Hermitian.

# 4 Functions and objects

QUANTY defines several functions and objects that can be used in the Lua-based script language. In this section, we discuss in more detail a few global functions and the objects `Wavefunctions`, `Operators`, `ResponseFunctions`, and `TightBinding`. Objects represent data structures and can come with a library of functions that are specific to the object type. An extended documentation of these functions and objects can be found at www.quanty.org [31]. Furthermore, the folder *TestScripts* in the codebase contains a separate file for each implemented function. Each test script provides an example of the function and tests its intended functionality.

## 4.1 Objects

In QUANTY, objects are represented as userdata type as defined in Lua. The implemented userdata type typically contains a pointer to a C structure with the data of the object. Userdata in Lua include meta-data that contain information on how the program should deal with operations on these objects such as addition and multiplication. In addition, user data types can be called, evaluated, and indexed. In the following, we will show, using several examples, how this is implemented and used in QUANTY.

### 4.1.1 Operator

Operators in QUANTY are defined in second quantization. The most basic operator that one can use is an operator that creates or annihilates a particle. For a basis with `NF=6` fermionic

states and `NB=0` bosonic states we can define the creation and annihilation operator on the one-particle state with index $i = 0$ as

```
1  NF=6
2  NB=0
3  OppAn = NewOperator("An",NF,0)
4  OppCr = NewOperator("Cr",NF,0)
```

You can add and multiply operators

```
5  OppN = OppCr * OppAn
```

You can compare two operators

```
6  OppNalso = 1 - OppAn * OppCr
7  print(OppNalso==OppN)
```

The previous script yields the output

```
true
```

As one can see from the previous example, the commutation relations between operators are implemented and used in the code. Operators can be printed to the standard output:

```
8  O = OppAn + 1.5*OppCr
9  O.Name = "My Operator Name"
10 print(O)
```

This generates the output

```
Operator: My Operator Name
QComplex       =           0 (Real==0 or Complex==1 or Mixed==2)
MaxLength      =           1 (largest number of products of lader ope
NFermionic modes =         6 (Number of fermionic modes (site, spin,
NBosonic modes   =         0 (Number of bosonic modes (phonon modes,

Operator of Length   1
QComplex       =           0 (Real==0 or Complex==1)
N              =           2 (number of operators of length   1)
A   0 |   1.00000000000000E+00
C   0 |   1.50000000000000E+00
```

The first line specifies the name of the operator. This can be any string and can be set by assigning the `Name` property of an operator object to a string. This has been done on line 9 of the previous example input. Lines 2 to 5 of the output give some information about the operator. The important parts are in the last two lines of the output. On these lines we see that the operator `O` consists of two terms (each on its own line) that annihilate a particle at site 0 with coefficient 1.0 or create a particle at site 0 with coefficient 1.5. If we change line 3 of the input to

```
3  OppAn = NewOperator("An",NF,1)
```

then the last two lines of the print output would be

```
A   1 |   1.00000000000000E+00
C   0 |   1.50000000000000E+00
```

One can save operators to disk with the command `io.put` and read them back with the command `io.get`. These commands expect two input variables. The first string specifies the file name where the object will be stored. The second string contains the actual name of the object. As an example, we can have a look at the following script.

```
11  io.put("filename","OppAn")
12  OppAn = nil
13  io.get("filename","OppAn")
```

The previous script in line 11 saves the operator `OppAn` in the file (folder) *filename*. In line 12 the operator is erased from memory and in line 13 the operator is retrieved from file. Multiple variables can be stored in the same file as long as they have different names within QUANTY.

One can generate sums over creation and annihilation operators with single short-hand commands. The operator

$$O = 10 + 0.1a_0^\dagger a_1^\dagger + (i+1)a_0^\dagger a_1^\dagger a_3 \tag{47}$$

can be created with the command

```
1  NF = 4 -- or larger depending on the size of the
2          -- one particle Hilbert space
3  NB = 0
4  O = NewOperator(NF, NB, {{            10},
5                           {0, 1,       0.1},
6                           {0,1,-3,     1+I}},
7                  {{"Name","Liberty"}})
```

The first two arguments of `NewOperator` indicate the dimension of the single-particle basis for fermions and bosons. The third argument contains a table. Each table element defines a term in the operator. Each term is defined by a table containing the indices where the particles are created (plus sign) or annihilated (minus sign) and a coefficient. An alternative way to define a one-particle operator is to start from a matrix. The input

```
1  M = {{1,0,0},
2       {0,1,0},
3       {0,0,1}}
4  O = Matrix.ToOperator(M)
```

creates the operator

$$O = \sum_{\tau_1,\tau_2=1,1}^{3,3} M_{\tau_1,\tau_2} a_{(\tau_1-1)}^\dagger a_{(\tau_1-1)}. \tag{48}$$

Note that matrices follow the Lua indexing starting at 1, whereas operators follow the C indexing starting at 0.

With the use of creation and annihilation operators, one can generate all possible operators. Although complete and possible, working on this level can be quite cumbersome. There are several standard operators implemented that allow one to generate operators quickly.

One set of standard operators is related to atomic physics. For an atomic sub-shell with angular momentum $l$ one can define several operators related to the spin ($S_x$, $S_y$, $S_z$, $S_+$, $S_-$, $S^2$), to the orbital angular momentum ($L_x$, $L_y$, $L_z$, $L_+$, $L_-$, $L^2$), to the total angular momentum ($J_x$, $J_y$, $J_z$, $J_+$, $J_-$, $J^2$), for spin-orbit coupling ($\sum_i \vec{l}_i \cdot \vec{s}_i$), for the Coulomb interaction ($\sum_{i,j>i} \frac{1}{|\vec{r}_i-\vec{r}_j|}$), or related to local (crystal field) potentials ($V(\vec{r})$).

The input for all of these operators follows the format

```
1  O = NewOperator("Type", NF, IndexUp, IndexDn)
```

or

```
1  O = NewOperator("Type", NF, IndexJmin, IndexJplus)
```

The string "Type" can be "Sx", "Sy", "Sz", "Splus", "Smin", "Ssqr", for one of the spin operators, or similar with "S" replaced by "L" for the orbital and replaced by "J" for the total angular

momentum. The operator for spin-orbit coupling is given by the "Type" string "ldots", for Coulomb interaction by the string "U", and for potentials (crystal fields) by the string "CF".

The labels `IndexUp` and `IndexDn` should be two lists of length $2l+1$ relating the Fermionic indices to the atomic states with spin up and spin down (quantized in the $z$ direction) with $m_l = -l$ to $m_l = l$. If one wants to work on a relativistic basis with $j$ coupled states, the indices `IndexJmin` and `IndexJplus` should be two lists of length $2l$ and $2l+2$ with $m_j = -l+1/2$ to $m_j = l-1/2$ for the list `IndexJmin` labeling the states with $j = l-1/2$ ($\kappa = j+1/2$) and with $m_j = -l-1/2$ to $m_j = l+1/2$ for the list `IndexJplus` labeling the states with $j = l+1/2$ ($\kappa = -(j+1/2)$).

The standard atomic sub-shell operators are only aware of the angular part of the wave function and need to be multiplied with parameters obtained from integrals over the radial wave function. For the spin-orbit coupling operator this is the spin-orbit coupling constant $\zeta$. For the Coulomb and crystal field operators, there are several integrals that are given as additional input parameters to the operator. The Coulomb operator for the interaction within one sub-shell with angular momentum $l$, expects $l+1$ Slater integrals as input. For a $d$-shell, one would need to provide 3 Slater integrals,

```
1  OppU = NewOperator("U", NF, IndexUp, IndexDn, {F0, F2, F4})
```

The values of the Slater integrals can be obtained from atomic radial functions or spherical averaged Wannier functions as for example obtained with FPLO or other Hartree-Fock or DFT codes. An example can be found in section 5. If the number of Slater integrals given is not in agreement with the total number expected, the code will give an error indicating the line number and some information on what went wrong. The Coulomb operator within a sub-shell of specific angular momentum needs one set of indices for spin up and down. The Coulomb operator between two sub-shells can be generated by providing two sets of indices for the spin up and down of the two sub-shells involved.

The crystal field operator represents a potential $V(\vec{r})$ acting within an atomic sub-shell with angular momentum $l$ or between two sub-shells with angular momentum $l_1$ and $l_2$. One can efficiently implement this operator by expanding the potential on spherical harmonics. The advantage of such an expansion is that the resulting angular integrals are analytically known, and the radial integrals become parameters that one either fits to experiment or calculates from a mean-field potential (DFT or Hartree-Fock) and the atomic radial functions.

A potential $V(\vec{r})$ expanded on spherical harmonics can be written in spherical coordinates as a Taylor expansion in $r$ times a sum over spherical harmonics in $\theta$ and $\varphi$

$$V(r,\theta,\varphi) = \sum_{k=0}^{\infty} \sum_{m=-k}^{k} \frac{1}{\sqrt{(k-m)!}} \frac{1}{\sqrt{(k+m)!}} \partial_z^{k-|m|}(-\text{Sign}[m]\partial_x + \iota\partial_y)^{|m|} V(r,\theta,\varphi)\Big|_{r=0}$$
$$\times r^k \sqrt{\frac{4\pi}{2k+1}} Y_{k,m}(\theta,\varphi). \tag{49}$$

The integral $\langle \phi_{\tau_1} | V | \phi_{\tau_2} \rangle$ can now be rewritten into a $r$ dependent part and a $\theta$ and $\varphi$ dependent part:

$$\langle \phi_{\tau_1} | V | \phi_{\tau_2} \rangle = \sum_{k=0}^{\infty} \sum_{m=-k}^{k} A_{k,m} \langle Y_{l_1,m_1} | C_{k,m} | Y_{l_2,m_2} \rangle, \tag{50}$$

with:

$$A_{k,m} = \frac{1}{\sqrt{(k-m)!}} \frac{1}{\sqrt{(k+m)!}} \langle R_{n_1,l_1} | r^k | R_{n_2,l_2} \rangle \left( \partial_z^{l-|m|}(-\text{Sign}[m]\partial_x + \iota\partial_y)^{|m|} V(r,\theta,\phi)\big|_{r=0} \right). \tag{51}$$

In crystal field theory $V(\vec{r})$ is an effective potential which is taken to fit the experiment, in practice one takes the values $A_{k,m}$ as the fitting parameters. The crystal-field Hamiltonian thus becomes:

$$H_{CF} = \sum_{\tau_1, \tau_2} \sum_{k,m} A_{k,m} \langle Y_{l_1,m_1} | C_{k,m} | Y_{l_2,m_2} \rangle a_{\tau_1}^\dagger a_{\tau_2}. \tag{52}$$

For most cases, one does not need all expansion coefficients $A_{k,m}$ of the potential. If symmetry is present, many of these coefficients are zero. Several other nonzero coefficients will be related to each other. Furthermore, if two shells with angular momentum $l_1$ and $l_2$ are coupled to each other, the value of $k$ must be in the range from $|l_1 - l_2|$ to $l_1 + l_1$ in steps of 2 such that $l_1 + l_2 + k$ is even.

A list of nonzero expansion coefficients for the different point groups and the relation of the expansion coefficients to the eigenenergies of an atomic, $s$-, $p$-, $d$- or $f$-shell can be found at https://www.quanty.org/physics_chemistry/point_groups [31].

In QUANTY one can create crystal field operators with the function "NewOperator()" and as a first input the string "CF". The function furthermore needs to know the effective potential expanded on renormalized spherical harmonics. This is given as a list of the form "$\{\{k_1, m_1, A_{k_1,m_1}\}, \{k_2, m_2, A_{k_2,m_2}\}, ...\}$". For the crystal field acting within a $d$-shell in cubic point group symmetry we have

```
-- crystal field operator in Oh
-- symmetry acting on a d-shell
NF = 10
NB = 0
IndexDn_3d = {0,2,4,6,8}
IndexUp_3d = {1,3,5,7,9}
tenDq = 1.1
Akm = {{4, 0,(21/10)*tenDq},
       {4,-4,(3/2)*math.sqrt(7/10)*tenDq},
       {4, 4,(3/2)*math.sqrt(7/10)*tenDq}}
OppCF = NewOperator("CF", NF, IndexUp_3d, IndexDn_3d, Akm)
```

In addition to the standard operators acting on a single atomic site, several operators and methods are available to create lattice models. A convenient way to generate operators for a lattice model with local interactions is by first defining a tight-binding model with periodic boundary conditions. One then can generate a super cell and from this create a finite-size lattice model. Details will be explained in the section on tight-binding objects 4.1.4.

### 4.1.2 Wavefunction

Wavefunction objects in QUANTY store many-body quantum states. These are given as sums of single Slater determinants as indicated in equation 9. For a state with $n$ electrons and $N_F$ basis states, there are $\binom{N_F}{n}$ possible Slater determinants. In QUANTY we do not store all $\binom{N_F}{n}$ coefficients but only those with a nonzero value. For example, a state on a one-particle basis describing a $p$-shell with a $p_x$ electron with spin down and a $p_y$ electron with spin up can be created with the command

```
NF = 6
NB = 0
BasisDn = {0,2,4}
BasisUp = {1,3,5}
psi = NewWavefunction(NF,NB,{{"110000", I/2},
                            {"010010", I/2},
                            {"100001", I/2},
                            {"000011",-I/2}})
```

In most calculations, wavefunctions are not created by hand, but as eigenstates of an operator. This can be done with the function `Eigensystem` and will be described in section 4.2.2 where we discuss different global functions.

In QUANTY, one can calculate the inner product of two functions using the function `Braket`. The innerproduct $S = \langle \psi_1 | \psi_2 \rangle$ is calculated as

```
1  S = Braket ( psi1 , psi2 ).
```

Acting with an operator on a state is done by multiplication, $|\psi_2\rangle = O |\psi_1\rangle$ is calculated as

```
1  psi2 = O * psi1
```

The state with a $p_x$ electron with spin down and a $p_y$ electron with spin up can also be created by defining two creation operators that act on a state with zero electrons.

```
1  NF = 6
2  NB = 0
3  BasisDn = {0,2,4}
4  BasisUp = {1,3,5}
5
6  -- define state psi0 with zero electrons
7  psi0 = NewWavefunction(NF, NB, {{"000000",1}})
8
9  -- define two operators that create an electron in the
10 -- p_x or p_y orbital with spin down or up respectively
11 CRpxdn = sqrt(1/2) *    ( NewOperator("Cr",NF,0)
12                         - NewOperator("Cr",NF,4) )
13 CRpyup = sqrt(1/2) * I * ( NewOperator("Cr",NF,1)
14                         + NewOperator("Cr",NF,5) )
15
16 -- define the same state as on line 5 of the previous example
17 psi = CRpxdn * CRpyup * psi0
```

### 4.1.3 Response Function

Response functions in QUANTY are implemented as sums over poles.

$$G(\omega, \Gamma) = A_0 + \sum_{i=0}^{N-1} B_i \frac{1}{\omega + i\Gamma/2 - a_{i+1}}, \tag{53}$$

with $A_0$ and $B_i$ $m$ by $m$ square matrices and $a_{i+1}$ real numbers. The matrix $B_i$ must be positive definite, and the matrix $A_0$ is zero for physical response functions, but nonzero for self energies. One normally would obtain a response function from the function `CreateSpectra`, or from a tight-binding object, but it can be defined by hand. A two dimensional response function with 3 poles at energies -1, 0 and 2 with residue $B_0$, $B_1$ and $B_2$ can be defined using the function `ResponseFunction.New`

```
1  A0 = Matrix.New({{0,0},{0,0}})
2  a1 = -1
3  a2 = 0
4  a3 = 2
5  B0 = Matrix.New({{1,0},{0,1}})
6  B1 = Matrix.New({{2,1},{1,2}})
7  B2 = Matrix.New({{2,0},{0,1}})
8  G  = ResponseFunction.New({ {A0,a1,a2,a3}, {B0,B1,B2},
9                              mu=0, type="ListOfPoles", name="G"})
```

Note that all $B_i$'s must be positive definite matrices with existing square root.

The response functions can be evaluated at arbitrary values of $\omega$ and $\Gamma$. For example

```
10  omega = 0.1
11  Gamma = 1.0
12  print(G(omega,Gamma))
```

returns

```
{ { (0.53819946 - 4.4476869 I) , (0.3846153 - 1.923076 I) } ,
  { (0.38461538 - 1.9230769 I) , (1.0304274 - 4.318153 I) } }
```

which is equal to

$$\begin{pmatrix} 1 & 0 \\ 0 & 1 \end{pmatrix} \frac{1}{\omega + i\Gamma/2 + 1} + \begin{pmatrix} 2 & 1 \\ 1 & 2 \end{pmatrix} \frac{1}{\omega + i\Gamma/2} + \begin{pmatrix} 2 & 0 \\ 0 & 1 \end{pmatrix} \frac{1}{\omega + i\Gamma/2 - 2}. \tag{54}$$

Responsefunctions can be stored and entered using different *types*. The type *ListOfPoles* stores responsefunctions as a sum over pole energies and their residues. One can also store responsefunctions using a block tri-diagonal representation, an Anderson representation or a natural impurity orbital representation. Transformations between these types can be made with the function `ResponseFunction.ChangeType`

One can add and subtract response functions from each other or multiply response functions with a constant. Note that the result of such operations must yield causal response functions. The weight or residue of all poles must be positive. Negative spectral weight is automatically removed from a pole and is either merged with poles at similar energy, or an error message is given. Given two response functions, one can calculate the self-energy. For example,

```
1  Sigma = ResponseFunction.CalculateSelfEnergy(G0,G)
```

calculate the self energy as

$$\Sigma(\omega) = G_0(\omega)^{-1} - G(\omega)^{-1}, \tag{55}$$

whereby $\Sigma(\omega)$, $G_0(\omega)$, and $G(\omega)$ are expressed as sums over poles. Furthermore, one can calculate the interacting bath Green's function defining the interacting hybridization function. This function is needed for dynamical mean field theory. The function

```
1  Gbath = ResponseFunction.CalculateHybridizationFunction(G0,Sigma)
```

calculates

$$G_{bath}(\omega) = \frac{1}{G_0(\omega - \Sigma(\omega))^{-1} + \Sigma(\omega)}. \tag{56}$$

For the case where the self-energy is fully local and thus $\vec{k}$ momentum independent, this yields the Green's function of a model where all sites except for the site where the Green's function is measured have a self-energy.

### 4.1.4  TightBinding

Many calculations for solids and molecules can be cast into the language of a tight-binding model. QUANTY allows one to define tight-binding Hamiltonians on an open or periodic lattice. Tight-binding objects are created with the function `NewTightBinding`. One can set the lattice constants, the atoms, and the hopping parameters between the atoms for different lattice vectors. The following script for example creates a tight-binding Hamilonian for a Dichalcogenide lattice

```
1  dAB = 0.2
2  tnn = 1.1
3  -- create the tight-binding Hamiltonian
4  HTB = NewTightBinding()
5  HTB.Name = "dichalcogenide tight-binding"
6  HTB.Cell = {{sqrt(3),0,0},
7              {sqrt(3/4),3/2,0},
8              {0,0,1}}
9  HTB.Atoms = { {"A", {0,0,0},         {{"p", {"0"}}}},
10             {"B", {sqrt(3),1,0}, {{"p", {"0"}}}}}
11 HTB.Hopping = { {"A.p","A.p",{        0,    0,0},{{-dAB/2}}},
12                 {"B.p","B.p",{        0,    0,0},{{ dAB/2}}},
13                 {"A.p","B.p",{        0,    1,0},{{ tnn   }}},
14                 {"B.p","A.p",{        0,   -1,0},{{ tnn   }}},
15                 {"A.p","B.p",{ sqrt(3/4),-1/2,0},{{ tnn   }}},
16                 {"B.p","A.p",{-sqrt(3/4), 1/2,0},{{ tnn   }}},
17                 {"A.p","B.p",{-sqrt(3/4),-1/2,0},{{ tnn   }}},
18                 {"B.p","A.p",{ sqrt(3/4), 1/2,0},{{ tnn   }}}
19                 }
```

Tight-binding objects can be used to calculate a band structure (`CalculateBands`) or density of states (`CalculateG`). Furthermore, one can transform tight-binding objects into operators. For this, one needs to define a set of atoms to include or a super cell with either open, or periodic boundary conditions. The example

```
10 HCl = CreateClusterHamiltonian(HTB, {"periodic",
11                                      {{4,0,0},{0,4,0},{0,0,1}}})
```

creates a cluster Hamiltonian with 16 sites in the ab plane. It is possible to read a tight-binding Hamiltonian from a Hartree-Fock or density functional theory calculation. We will discuss this in section 5.1 by the example of NiO.

## 4.2 Global functions

QUANTY implements several global functions: `BlockBandDiagonalize`, `Braket`, `BraketDiagonal`, `CalculateBands`, `CalculateDOS`, `CalculateFermi`, `CalculateG`, `CalculateRho`, `Chop`, `Clone`, `Conjugate`, `ConjugateTranspose`, `Copy`, `CreateAtomicIndicesDict`, `CreateAtomicIndicesList`, `CreateSpectra`, `CreateFluorescenceYield`, `CreateResonantSpectra`, `CreateClusterHamiltonian`, `DensityMatrix`, `DeterminantString`, `Eigensystem`, `Expand`, `ExpandToBasis`, `FileReadStuttgartCTRL`, `FileReadStuttgartHAMR`, `FileReadDresdenFPLO`, `FindAllAtomsInsideSphere`, `GetSlaterIntegrals`, `GetMultipoleBesselIntegral`, `GetMultipoleIntegral`, `HybridizationFunctionFromFPLO`, `ImpurityGreensFunction`, `Inverse`, `MeanFieldGroundState`, `MeanFieldOperator`, `NewMatrix`, `NewOperator`, `NewResponseFunction`, `NewTightBinding`, `NewWavefunction`, `NonRelToRelOrbitals`, `OperatorToMatrix`, `OperatorSetOnsiteEnergy`, `OperatorSetTrace`, `OrbitalRotationMatrix`, `OrbToMultiplicity`, `Orthonormalize`, `PartialMeanFieldOperator`, `PartialOperator`, `PotentialExpandedOnClm`, `PlotBands`, `PlotBandsAndDOS`, `PlotDOS`, `PlotFermi`, `ReadFPLO`, `PrintExpectationValues`, `ReadFPLOBasisFunctions`, `ReadLenaBasisFunctions`, `RelToNonRelOrbitals`, `Rotate`, `Statistics`, `TensorProduct`, `TimeEnd`, `TimePrint`, `TimeSet`, `TimeStart`, `TightBindingDefFromStuttgartCTRLHAMR`, `TightBindingDefFromDresdenFPLO`, `Transpose`, `Verbosity`, `YtojjzMatrix`, `YtoKMatrix`, `YtoZMatrix`, `ZtoKMatrix`. Detailed documentation of these functions can be found at www.quanty.org [31]. In the folder *TestScripts* of the code base one can find for each of these functions a set of examples with the output they produce.

In the following, we highlight a few functions that form the core of many calculations.

### 4.2.1 Braket

The function `Braket` can be used to calculate the inner product of states or the expectation values of operators. The input

```
1  S = Braket(Psi1,Psi2)
```

calculates

$$S = \langle \Psi_1 | \Psi_2 \rangle \tag{57}$$

$$= \int_{\mathbb{R}^3} \dots \int_{\mathbb{R}^3} \Psi_1^*(\vec{r}_1, \dots, \vec{r}_n) \Psi_2(\vec{r}_1, \dots, \vec{r}_n) \, d\vec{r}_1 \dots d\vec{r}_n$$

Similar expectation values of operators are calculated as

```
1  O12 = Braket(Psi1,O,Psi2)
```

which calculates

$$O_{1,2} = \langle \Psi_1 | O | \Psi_2 \rangle. \tag{58}$$

The function `Braket` expands over lists of functions such that

```
1  Slist = Braket(Psi1,{PsiA,PsiB,PsiC})
```

is equivalent to

```
1  Slist = {Braket(Psi1,PsiA),Braket(Psi1,PsiB),Braket(Psi1,PsiC)}
```

The function `Braket`(Psi1,Psi2) is equivalent to `Psi1*Psi2`, but especially when treated over lists of functions with many determinants, it can be much faster.

### 4.2.2 Eigensystem

The function `Eigensystem` calculates the lowest $N$ eigenstates of an operator. Internally different algorithms are used depending on the properties of the problem. The general idea of QUANTY is that we never represent operators as matrices, but use iterative Lanczos methods. We define a set of (random) starting vectors and iteratively act with the Hamiltonian on these states to find states with lower energy. Implemented are shifted, restarted, normal and block, Lanczos routines. The convergence criterion is given by the variance of the energy. The variance should be zero for an exact eigenstate and can be required to be small for a numerical solution. Within QUANTY we run a standard Lanczos loop to find the $N$ lowest eigenstates. We calculate $N_{\text{tri}}$ Krylov basis states and use these to diagonalize the Hamiltonian. If the variance of one of the lowest $N$ eigenstates within the Krylov basis is larger than $\epsilon$ we restart a Lanczos loop, taking the previous result as a starting point. The value for the convergence criteria, i.e. $\epsilon$, is standard set at $\sqrt{\text{DBL\_EPSILON}}$ but can be changed to a user defined value.

In order for this algorithm to run, one needs to define a starting point. The starting point can be a random state in the full Hilbert space. We often can restrict the possible random states to a starting point that is closer to the ground state. We can either define the starting point using a set of starting states or by setting restrictions on the possible states.

If `PsiList` is a list that contains a set of $N$ states, then

```
1  Eigensystem(Hamiltonian,PsiList)
```

will find the $N$ lowest eigenstates of the operator `Hamiltonian` that have a nonzero overlap with at least one of the states in `PsiList`.

Using restrictions needs a bit more explanation; it implements a very flexible way to do several restrictive active-space calculations. Given a one-particle basis with $N_F$ one-particle states, we can now define a set of restrictions that list the minimum and maximum number

of one-particle states that can be occupied. For each restriction, we use a bit field of length $N_F$ that tells the code if a single particle state is included or is not included in the counting. For example if one has two $p$-shells with in total six electrons and the first should be roughly occupied by 2 and the second by roughly 4 electrons, one can set the starting restrictions to

```
1  NF = 12
2  StartRestrictions = {NF, NB,  {"111111 000000",2,2},
3                                 {"000000 111111",4,4}}
```

Calculating the `Npsi` lowest eigenstates can be done with the function

```
4  PsiList = Eigensystem(Hamiltonian, StartRestrictions, Npsi)
```

At the beginning a set of single Slater determinant states in the $p^2 p^4$ configuration will be generated. If the Hamiltonian contains terms that allow electrons to hop between the two $p$-shells then during the iterative procedure in the first step additional states in the $p^3 p^3$ and $p^1 p^5$ configurations will be generated. In the next iterative step, additional states in the $p^4 p^2$ and $p^0 p^6$ configurations will be generated. In the third step, states in the $p^5 p^1$ and in the fourth step states in the $p^6 p^0$ configuration will be generated. After 4 iterations, the complete Hilbert space for 6 total electrons is probed. One can restrict the configurations included during the convergence cycles. This is done by setting restrictions in the options of the function `Eigensystem`.

If one only wants to include the configurations $p^2 p^4$ and $p^3 p^3$, one can restrict the occupation of the first $p$-shell between 2 and 3. The code to do this is

```
1  NF = 12
2  StartRestrictions       = {NF, NB,  {"111111 000000",2,2},
3                                       {"000000 111111",4,4}}
4  CalculationRestrictions = {NF, NB,  {"111111 000000",2,3}}
5  PsiList = Eigensystem(Hamiltonian, StartRestrictions, Npsi,
6                   {{"Restrictions",CalculationRestrictions}})
```

Whereby we still start with random states taken from the $p^2 p^4$ configuration, and during the expansion only allow states in the $p^2 p^4$ and $p^3 p^3$ configurations. Different restrictions can be combined, allowing for a flexible definition of restricted active spaces and truncated configuration interaction schemes.

### 4.2.3 CreateSpectra

Given a state $\Psi$ an operator $H$ and a transition operator $T$ the spectral function $G(\omega) = \left\langle \Psi \left| T^\dagger \frac{1}{\omega - H + E_0 + i\Gamma/2} T \right| \Psi \right\rangle$, with $E_0 = \langle \Psi | H | \Psi \rangle$ can be calculated with the command

```
1  S, G = CreateSpectra(H, T, Psi)
```

The variable $S$ contains a spectrum object. Spectra objects contain a list of energies, and real and imaginary parts of the response function at that energy. The energy grid can be given as an option in the function `CreateSpectra`. When absent, reasonable values are determined automatically. $G$ contains the same information, but as a response function object. See Section 4.1.3 on response functions for more information.

The function `CreateSpectra` starts with the state $T | \Psi \rangle$. Then it builds a Krylov basis for this state spanned by the states $H^n T | \Psi \rangle$, with $n \in [0, \ldots, N_{Tri} - 1]$. The variable $N_{Tri}$ can be set as an option in the function `CreateSpectra` and is set standard at 200. Within the Krylov subspace, the response function is calculated.

Similarly to the eigenstate calculations, one can restrict the configurations included in the calculation of the spectra. The format is the same as for the function Eigensystem.

If one provides a list of functions and or a list of operators, a set of response functions is created. One can create a tensor with the option {"Tensor",true}. For example, if $T_x$, $T_y$ and

$T_z$ are the transition operators for polarized light parallel to the $x$, $y$ and $z$ directions, then the command,

```
1  S, G = CreateSpectra(H, {Tx,Ty,Tz}, Psi,{{"Tensor",true}})
```

will generate the 3 by 3 conductivity tensor of the system that can be used to calculate the spectrum for any polarization. Note that in this case, the spectrum object $S$ will have 9 columns of information, and the response function object $G$ will be represented by a 3 by 3 matrix, i.e., it is given by a sum over poles with a 3 by 3 matrix as residue.

### 4.2.4  CreateResonantSpectra

The function `CreateResonantSpectra` allows one to calculate higher-order response functions. The function

```
1  S = CreateResonantSpectra(H1 , H2 , T1 , T2 , Psi)
```

creates the spectral function

$$S = \left\langle \Psi \left| T_1^\dagger \frac{1}{\omega_1 - H_1 + E_1 - i\frac{\Gamma_1}{2}} T_2^\dagger \frac{1}{\omega_2 - H_2 + E_2 - i\frac{\Gamma_2}{2}} T_2 \frac{1}{\omega_1 - H_1 + E_1 + i\frac{\Gamma_1}{2}} T_1 \right| \Psi \right\rangle, \quad (59)$$

with $E_1 = \langle \Psi | H_1 | \Psi \rangle$ and $E_2 = \langle \Psi | H_2 | \Psi \rangle$. The boundaries for $\omega_1$ and $\omega_2$ as well as $\Gamma_1$ and $\Gamma_2$ and the number of discrete energy points for $\omega_1$ and $\omega_2$ can be given as options. If the range for $\omega_1$ or $\omega_2$ is not given, reasonable values will be assumed for the calculation at hand. Similarly to the function `CreateSpectra` and `Eigensystem`, one can set restrictions on the configurations included for the calculation. It is possible to set different restrictions for the expansion in $H_1$ and $H_2$.

### 4.2.5  Rotate

One useful function that is used to optimize calculations or to bridge between different codes is the function `Rotate`. The function `Rotate` can be used to make (unitary) transformations of the underlying one-particle basis states. Rotate works on operators, states, response functions, and tight-binding objects. For example, we can rotate a state to reduce the number of determinants needed to represent the state:

```
1  NF=2
2  NB=0
3  Psi0=NewWavefunction(NF, NB, { { "10",sqrt(1/2)}, {"01",sqrt(1/2)} } )
4   u = {{ sqrt(1/2),sqrt(1/2)},
5        {-sqrt(1/2),sqrt(1/2)}}
6  PsiR=Rotate(Psi0,u)
```

In the previous example, the state $|\Psi_R\rangle$ contains one determinant and is given as $|\Psi_R\rangle = a_0^\dagger |0\rangle$.

Rotate works on operators and on states. We rotate operators and states such that given a unitary matrix $u$, and

```
1  PsiR  =  Rotate(Psi,u)
2  OR    =  Rotate(O,u)
```

then

$$\langle \Psi_R | O_R | \Psi_R \rangle = \langle \Psi | O | \Psi \rangle. \quad (60)$$

Furthermore, when $O$ is a one-particle operator $O = \sum_{\tau_1,\tau_2} o_{\tau_1,\tau_2} a_{\tau_1}^\dagger a_{\tau_2}$ then the following script generates a diagonal operator $OR$

```
1  M = OperatorToMatrix(O)
2  val, fun = Eigensystem(M)
3  OR = Rotate(O,fun)
```

with $OR = \sum_\tau or_\tau a_\tau^\dagger a_\tau$.

# 5   Examples

The codebase contains several test scripts (in the folder *TestScripts*) that call each function and compare the results to the expected result. These tests are automatically run for each new commit to the git repository. In addition, several examples and tutorials are available on the QUANTY website [31].

In this section, we present an example of how to calculate several types of x-ray spectroscopy. More specifically, we show an example of a density functional theory-based ligand field theory calculation of NiO [29]. We divide the example into several subsections that indicate the basic idea of how to build a calculation script in QUANTY. The first subsection reads the parameters of a ligand field theory calculation from a converged DFT calculation. The next subsection shows how to calculate the ground state. The final subsections calculate several spectroscopic properties of NiO.

NiO has 8 electrons in the $d$-shell and a filled O-$2p$ derived band. On a DFT level NiO is a metal. In reality, NiO becomes an insulator due to strong Coulomb interactions within the Ni $3d$ orbitals. As such, NiO is a Mott-Hubbard insulator. In order to explain the optical and photoelectron gap one needs to include the O-$2p$ derived bands. The lowest photoelectron spectroscopy state has holes in the oxygen bands, which is why the size of the band gap is smaller than the size of the local Hubbard $U$ [85]. As such, NiO is classified as a charge transfer insulator. Within ligand field theory we calculate the properties of the solid from a $NiO_6$ cluster. The tight-binding Hamiltonian is generated from the Wannier orbitals corresponding to the Ni-$3d$ and O-$2p$ orbitals from DFT calculation. The full Coulomb repulsion and spin-orbit coupling within the Ni-$3d$ are added to the Hamiltonian.

## 5.1   Reading the output of a converged DFT calculation for NiO

In QUANTY one can read the tight-binding structure and hopping parameters of local Wannier orbitals of a converged DFT calculation from the code FPLO [86]. The following script first reads all the output from a converged DFT calculation into the variable `FPLOOut`. Then it generates a tight-binding object from the output and stores this in the variable `TB`.

```
1  FPLOOut = FileReadDresdenFPLO("DFT/out.wan")
2  TB = TightBindingDefFromDresdenFPLO(FPLOOut)
```

Tight-binding objects can be used in different ways in the rest of the code.

In addition to the one-particle interactions, we also need the two-particle interactions. These are the Coulomb interactions between the electrons in the $d$-shell. We can again obtain these from the DFT orbitals. The Coulomb Hamiltonian on a $3d$-shell is defined by 3 parameters, the Slater integrals.

```
3  -- Calculating slater integrals from DFT radial wavefunctions
4  radialFunctions = ReadFPLOBasisFunctions({"3d"},"DFT/+fval.001.1")
5  slaterIntegrals = GetSlaterIntegrals({"3d"},radialFunctions)
```

In line 4 we read the atomic basis functions from DFT. We only include the Ni $3d$ orbital. In line 5 we calculate the Slater integrals from the radial wave functions. The variable `slaterIntegrals` contains a list with all available Slater integrals for the atoms and shells included in the input. In this case $F_0$, $F_2$, and $F_4$ are calculated for the Ni $3d$-shell using the atomic Ni DFT basis functions.

## 5.2   Ligand field theory groundstate

After reading the tight-binding Hamiltonian from DFT we can generate a $NiO_6$ cluster. This can be done with the following script.

```
6  -- Genreating the tigh-binding Hamiltonian
7  NewCluster = FindAllAtomsInsideSphere(TB.Atoms,TB.Cell,{0,0,0},4)
8  HDFTLarge, ClusterTB, IndexName = CreateClusterHamiltonian(TB,
9  {"open", NewCluster},  {{"AddSpin",false},
10 {"ReturnTBIndicesDict",true}})
11 RTB, T = BlockBandDiagonalize(ClusterTB,{{"Ni",{0,0,0}}})
12 HDFT, ClusterTB, IndexFPLO = CreateClusterHamiltonian(RTB,
13 {"open", RTB.Atoms}, {{"AddSpin",true}, {"ReturnTBIndicesDict",true}})
```

The function `FindAllAtomsInsideSphere`(TB.Atoms,TB.Cell,{0,0,0},r) finds all atoms within a sphere of radius r around the origin. This list is passed, on line 5 of the script, to the function `CreateClusterHamiltonian`. This function generates an operator in second quantization and a tight-binding object representing the same cluster. The variable `IndexName` stores information on the relation between the indices in the operator and the positions of the atom and the shell in the tight-binding object. Within a $NiO_6$ cluster, there are 18 O $2p$ orbitals. These can be linearly combined to 5 ligand orbitals that couple to the Ni-$3d$ orbitals, whilst the others have no direct interaction with the Ni $3d$ orbitals. This helps to reduce the size of the basis, and for this purpose, `BlockBandDiagonalize` in line 6 is used, which outputs the tight-binding object of the system with rotated orbitals (RTB). In line 7 the final one-particle part of the DFT based ligand field Hamiltonian is generated.

We can now add the Coulomb interaction to the Hamiltonian. We normally fit $F_0$ or $U$, the monopole part of the Coulomb interaction, to experiment and take $F_2$ and $F_4$ from DFT.

```
14 Udd      =  7.30
15 slaterIntegrals = GetSlaterIntegrals({"3d"},radialFunctions)
16 F2dd = slaterIntegrals["3d 3d 3d 3d"][2] * EnergyUnits.Ha.value
17 F4dd = slaterIntegrals["3d 3d 3d 3d"][4] * EnergyUnits.Ha.value
18 F0dd = Udd+(F2dd+F4dd)*2/63
19 OppU = NewOperator("U",10,Index["Ni_3d_Up"],Index["Ni_3d_Dn"],
20                                            {F0dd,F2dd,F4dd})
21 OppU = Rotate(OppU,YtoZMatrix("d"))
```

The Slater integrals in the function `GetSlaterIntegrals` are calculated in Hartree units. We multiply them with the constant `EnergyUnits.Ha.value` to transform them to eV. On line 18 we define the Coulomb operator using a standard function for operators. Note that QUANTY standard defines the $d$ orbital basis as complex spherical harmonics. FPLO works on a basis of the real tesseral harmonics. We thus need to bring the Coulomb and tight-binding operators to the same one-particle basis. We do this in line 20 of the script by rotating the one-particle basis, in this case of the Coulomb operator, from a basis of complex spherical harmonics to real tesseral harmonics.

In a DFT calculation the local Coulomb interaction between the $d$ electrons is included on a mean-field level. If we want to add the full Coulomb interaction, one needs to subtract the mean-field contribution. We can calculate a mean-field approximation of an operator with the function `MeanFieldOperator`. This function expects two inputs, an operator and a density matrix. For the example of NiO the mean-field operator is constructed as follows.

```
22 -- Generating mean-field Coulomb operator
23 rhoNoSpin = Chop(CalculateRho(TB))
24 rhoNoSpind = Matrix.Sub(rhoNoSpin,5)
25 rhod = Matrix.AddSpin(rhoNoSpind)
26 OppUMF = Chop(MeanFieldOperator(OppU,rhod))
```

The function `CalculateRho`(TB) calculates the mean-field density as a matrix on the basis of the FPLO tesseral harmonic basis orbitals. The Hartree-Fock approximation of the operator `OppU` is generated using `MeanFieldOperator`. The output is the Coulomb operator in the mean-field approximation, which will be subtracted from `OppU` to correct for double counting.

The full Hamiltonian consists of the DFT Hamiltonian plus Coulomb and spin-orbit interactions. A small magnetic field along the z-axis lifts the degeneracy of degenerate eigenstates. With `Eigensystem` (see Sec. 4.2.2), the lowest 3 eigenstates are calculated.

```
27  Hamiltonian = HDFT + OppU - OppUMF + Bz * (2*OppSz + OppLz)
28  + zeta_3d * Oppldots
29
30  Npsi=3
31  StartRestrictions = {NF, 0,
32                          {DeterminantString(NF,Index["Ni_3d"]),nd,nd},
33                          {DeterminantString(NF,Index["Ligand_d"]),10,10}}
34  psiList = Eigensystem(Hamiltonian, StartRestrictions, Npsi)
```

In line 30 we define a set of restrictions for the starting point of the iterative procedure of finding eigenstates. These are needed as an operator in second quantization is not defined for a specific configuration. For NiO we include the configurations $d^8 L^{10}$, $d^9 L^9$, and $d^{10} L^8$. After calculating the eigenstates it is useful to have a look at several expectation values.

```
35  oppList={Hamiltonian, OppSsqr, OppLsqr, OppJsqr, OppSz, OppLz,
36  Oppldots, OppN_Ni_eg, OppN_Ni_t2g, OppN_Ni, OppN_Ligand}
37  PrintExpectationValues(psiList,oppList,{{"ColWidth",10}})
```

The eigenenergies and expectation values with respect to standard operators can be calculated and printed using `PrintExpectationValues`. The result of the previous script is:

|   | <E> | <S^2> | <L^2> | <J^2> | <S_z> | <L_z> | <l.s> | <Neg> | <Nt2g | <N_Ni | <N_L> |
|---|-----|-------|-------|-------|-------|-------|-------|-------|-------|-------|-------|
| 1 | -3.4 | 1.77 | 10.9 | 13.8 | -0.9 | -0.3 | -0.34 | 2.2 | 5.99 | 8.19 | 9.81 |
| 2 | -3.4 | 1.77 | 10.9 | 13.8 | -5e-7 | -4e-5 | -0.34 | 2.2 | 5.99 | 8.19 | 9.81 |
| 3 | -3.4 | 1.77 | 10.9 | 13.8 | 0.9 | 0.3 | -0.34 | 2.2 | 5.99 | 8.19 | 9.81 |

## 5.3 X-ray Absorption Spectroscopy

After calculating the ground-state one can calculate different spectral functions. In this section, we will have a look at the X-ray absorption spectra of the $L_{2,3}$ edges in NiO. For this we can reuse the previous script with some small modifications. In order to calculate the $L_{2,3}$ edges, we need to expand the one-particle basis to include, besides the Ni 3d- and Ligand $Ld$-shell, also the Ni 2p-shell. For this, we change line 3 to 5 of the script that calculated the ground-state. In addition to the Ni 3d radial function, it should also read the Ni 2p radial function.

```
3  radialFunctions = { ReadFPLOBasisFunctions({"2p"},"DFT/+fcor.001.1")
4                    ,ReadFPLOBasisFunctions({"3d"},"DFT/+fval.001.1")}
5  slaterIntegrals = GetSlaterIntegrals({"2p","3d"},radialFunctions)
```

Note that the Ni-2p orbitals are in the core and the Ni-3d orbitals in the valence, which are stored in different files in the DFT output. Once we add the 2p orbitals to the basis, we also need to change the restrictions used to calculate the ground-state. Line 30 becomes

```
30  StartRestrictions = {NF, 0,
31                          {DeterminantString(NF,Index["Ni_3d"]),nd,nd},
32                          {DeterminantString(NF,Index["Ni_2pd"]),6,6},
33                          {DeterminantString(NF,Index["Ligand_d"]),10,10}}
```

For Ni-2p electrons we need to add the Coulomb interaction with Ni-3d electrons. The following script calculates these Coulomb matrix elements.

```
38  Upd      =   8.5
39  YtoZ_dLp =   YtoZMatrix({"Ni_3d","Ligand_d","Ni_2p"})
40  F2pd   =   slaterIntegrals["3d 2p 3d 2p"][2] * EnergyUnits.Ha.value
41  G1pd   =   slaterIntegrals["2p 3d 3d 2p"][1] * EnergyUnits.Ha.value
42  G3pd   =   slaterIntegrals["2p 3d 3d 2p"][3] * EnergyUnits.Ha.value
43  F0pd   = Upd + (1/15)*G1pd + (3/70)*G3pd
```

```
44 OppUpd= NewOperator("U",NF,Index["Ni_2p_Up"],Index["Ni_2p_Dn"],
45                     Index["Ni_3d_Up"],Index["Ni_3d_Dn"],
46                     {F0pd,F2pd}, {G1pd,G3pd})
47 OppUpd= Rotate(OppUpd,YtoZ_dLp)
```

In order to calculate spectroscopic excitations, we need a transition operator. For light-induced transitions these are given by the electric field of the light. After a multipole expansion of the light field, we can use the crystal field operator to implement dipole transitions.

```
48 t=math.sqrt(1/2)
49 Akm = {{1,-1,t},{1, 1,-t}}
50 TXASx = NewOperator("CF", NF, Index["Ni_3d_Up"],
51       Index["Ni_3d_Dn"], Index["Ni_2p_Up"], Index["Ni_2p_Dn"], Akm)
52 TXASx = Rotate(TXASx,YtoZ_dLp)
53 Akm = {{1,-1,t*I},{1, 1,t*I}}
54 TXASy = NewOperator("CF", NF, Index["Ni_3d_Up"],
55       Index["Ni_3d_Dn"], Index["Ni_2p_Up"], Index["Ni_2p_Dn"], Akm)
56 TXASy = Rotate(TXASy,YtoZ_dLp)
57 Akm = {{1,0,1}}
58 TXASz = NewOperator("CF", NF, Index["Ni_3d_Up"],
59       Index["Ni_3d_Dn"], Index["Ni_2p_Up"], Index["Ni_2p_Dn"], Akm)
60 TXASz = Rotate(TXASz,YtoZ_dLp)
61 TXASr = t*(TXASx - I * TXASy)
62 TXASl =-t*(TXASx + I * TXASy)
```

The X-ray transition operators are calculated using the crystal field operator (see section 4.1.1). In lines 61 and 62, right and left circularly polarized transition operators are defined.

Using the function `CreateSpectra` (see Sec. 4.2.3), the X-ray absorption spectra are calculated. As input, 3 transition operators and the lowest 3 eigenstates are given, so that 9 spectra are calculated.

```
63 Hamiltonian = HDFT + OppUdd - OppUddMF + OppUpd - OppUpdMF
64 + Bz * (2*OppSz + OppLz) + zeta_3d * Oppldots_3d
65 + zeta_2p * Oppldots_2p
66 XASSpectra = CreateSpectra(Hamiltonian, {TXASz, TXASr, TXASl},
67 psiList, {{"Emin",-15}, {"Emax",25}, {"NE",2000}, {"Gamma",0.1}})
68 XASSpectra.Print({{"file","XASSpec.dat"}})
```

On line 68 we print the real and imaginary part of the spectrum to an ASCII file on disc. This allows one to compare the calculated spectra with the experimental results [29, 87].

## 5.4  Core level Photo-electron Spectroscopy

It is relatively straightforward to examine different types of spectroscopy using QUANTY. Here we show the example of $2p$ x-ray photoemission spectra of Ni in NiO. For this type of spectroscopy, an electron from the $2p$-shell is excited by a photon to a free electron. We can think of these as plane waves, but equally well as spherical waves described by spherical harmonics times a Bessel function. For the latter, it becomes clear that within the dipole approximation the angular momentum of the free electron $l_{elec}$ is either 0 ($s$ orbital) or 2 ($d$ orbital). The ratio between these two channels depends on the photon energy. It is important to realize that in photoelectron spectroscopy there is a strong dependence of the intensity on the polarization of the light and the direction of the detector. These effects are often called matrix element effects. Within QUANTY it is relatively straight forward to include these [27, 28, 78–80, 88].

In the following script we assume a global Cartesian coordinate system. Within this coordinate system, we can represent the polarization of the light by a unit vector $\hat{\epsilon}$ and the direction of the detector with the two angles $(\theta_{detect}, \varphi_{detect})$. In this way $\theta$ defines the angle between the $z$ axis and the vector from the sample to the detector and $\varphi$ the angle between the $x$ axis and the projection of the vector between the sample and the detector onto the $z = 0$

plane. We can now define the following functions to generate the polarization- and detector position-dependent transition operators.

```
1  function YlmClmYlm (l1,m1,l2,m2,l3,m3)
2    return (-1)^m1 * sqrt((2*l1+1)*(2*l3+1))
3                   * ThreeJSymbol({l1,0},{l2,0},{l3,0})
4                   * ThreeJSymbol({l1,-m1},{l2,m2},{l3,m3})
5  end
6
7  function sqr(x)
8    return x*x
9  end
10
11 function TransitionOperator(l3, IndexDn, IndexUp, NF,
12                             epsilon_x, epsilon_y, epsilon_z,
13                             detector_theta, detector_phi)
14   local T = 0
15   local epsilon={}
16   epsilon[-1] = sqrt(1/2)*( epsilon_x - I*epsilon_y)
17     / sqrt(sqr(epsilon_x) + sqr(epsilon_y) + sqr(epsilon_z))
18   epsilon[ 0] =   epsilon_z
19     / sqrt(sqr(epsilon_x) + sqr(epsilon_y) + sqr(epsilon_z))
20   epsilon[ 1] = sqrt(1/2)*(-epsilon_x - I*epsilon_y)
21     / sqrt(sqr(epsilon_x) + sqr(epsilon_y) + sqr(epsilon_z))
22   -- T = sum_{m,m',m''} a_m <m|C_epsilon_m'|s> Y(lm'',theta,phi)
23   for m1=-1,1 do       -- angular momentum of the p electron
24     for m2=-1,1 do     -- angular momentum of the dipole operator
25       for m3=-l3,l3 do -- angular momentum of the free electron
26         T=T+NewOperator("An", NF, {IndexDn[m1+2]},
27             {epsilon[m2] * YlmClmYlm(1,m1,1,m2,l3,m3)*
28               SphericalHarmonicY(l3,m3,detector_theta,detector_phi)})
29           +NewOperator("An", NF, {IndexUp[m1+2]},
30             {epsilon[m2] * YlmClmYlm(1,m1,1,m2,l3,m3)*
31               SphericalHarmonicY(l3,m3,detector_theta,detector_phi)})
32       end
33     end
34   end
35   return Chop(T)
36 end
```

We here use the fact that in Lua one can define functions. The function `TransitionOperator` returns the photoemission transition operator and expects as input the angular momentum of the free electron, $l_{elec}$, the indices of the core states, the light polarization $\hat{\epsilon}$, and the detector angles ($\theta_{detect}$, and $\phi_{detect}$).

The script to calculate the spectra follows the example in the previous section on calculating the x-ray absorption spectra. The difference between the x-ray absorption and photoelectron spectroscopy script is a different transition operator $T$.

```
37 TPES_s = TransitionOperator (0, IndexDn_2p, IndexUp_2p, NF,
38 epsilon[1], epsilon[2], epsilon[3], detector_theta, detector_phi)
39 TPES_d = TransitionOperator (2, IndexDn_2p, IndexUp_2p, NF,
40 epsilon[1], epsilon[2], epsilon[3], detector_theta, detector_phi)
41 TPES = TPES_s + TPES_d
42 cPESSpectra = CreateSpectra(cPESHamiltonian, {TPES_s,TPES_d,TPES},
43 psiList, {{"Emin",-40}, {"Emax",10}, {"NE",5000}, {"Gamma",0.1}})
```

The results of core-level photoelectron spectroscopy calculations of NiO can be compared with experiments [29, 89]. Note that ligand field theory is not sufficient to capture the core-level photoelectron spectroscopy of NiO. Ligand field calculations agree well with experiments of Ni impurities in MgO, but show clear deviations from the spectra measured for bulk NiO [90]. The reason for these deviations are metal-to-metal or Mott-Hubbard charge transfer excita-

tions [91]. In principle these effects can be captured using dynamical mean field theory, but one has to be very careful on how to implement these equations. Several different DMFT implementations for the $2p$ core-level photoelectron spectra of NiO yield modest but significant different spectroscopic results. The differences are related to the details of the numerical solvers, the implementation of the Coulomb operator, the assumed magnetic order, and the accuracy of the crystal field parameters obtained from DFT [42, 90, 92–97]. Although very encouraging, these results show the importance of developing methods to determine reliable values for the screened interactions and to correct for possible double-counting of the interactions. At the same time, without reliable methods to obtain *ab initio* parameters underlying many-body calculations, it is useful to have a script language that allows one to modify the model parameters to fit the experiment, thus obtaining information on the validity of the model and the experimentally realized parameter values.

## 5.5 Non-resonant Inelastic X-ray Scattering

This example calculates the $d - d$ excitations in NiO using non-resonant Inelastic X-ray Scattering (nIXS) [3]. This is one of a spectroscopy technique with very straight forward selection rules. We use the $A(r, t)^2$ term of the light-matter interaction to make transitions between states. This is the leading term if one uses photons of much higher energy than the transition energies involved. These photons now carry non negligible momentum and one can make transitions beyond the dipole limit. More precise, by changing the scattering angle one can select the order of the multipole one measures. In this example we look at angular-momentum transfer of $k = 2$ and $k = 4$ for the transitions between the Ni-$3d$ orbitals. Creating the Hamiltonian and calculating the eigenstates is just as in the previous example discussed in section 5.2.

In order to calculate nIXS, we need to determine the intensity ratio for the different multipole intensities (see Ref. [3]). The $A(r, t)^2 \propto e^{iq \cdot r}$ interaction, where $q$ is the scattering momentum, is expanded on spherical harmonics and Bessel functions. The $3d$ Wannier functions are expanded on spherical harmonics multiplied by a radial wave function. The angular part of the integration of the transition strength can be calculated analytically. For the radial part, we calculate $\langle R_{3d}(r)|j_k(qr)|R_{3d}(r)\rangle$, where $R_{3d}(r)$ is the $3d$ atomic radial function and $j_k(qr)$ are spherical Bessel functions of order $k$. The following script can be used to calculate these integrals.

```
1  -- <R_3d(r) | j_k(q r) | R_3d(r)>
2  function RjRdd (q, R)
3      local r_arr = R.x
4
5      local bessel0 = {}
6      local bessel2 = {}
7      local bessel4 = {}
8      for ir, r in pairs(r_arr) do
9          bessel0[ir] = SphericalBesselJ(0,q*r)
10         bessel2[ir] = SphericalBesselJ(2,q*r)
11         bessel4[ir] = SphericalBesselJ(4,q*r)
12     end
13     bessel0 = InterpolatingFunction.Spline(r_arr, bessel0)
14     bessel2 = InterpolatingFunction.Spline(r_arr, bessel2)
15     bessel4 = InterpolatingFunction.Spline(r_arr, bessel4)
16
17     local Rj0R = InterpolatingFunction.Integrate(R*bessel0*R)
18     local Rj2R = InterpolatingFunction.Integrate(R*bessel2*R)
19     local Rj4R = InterpolatingFunction.Integrate(R*bessel4*R)
20
21     return Rj0R, Rj2R, Rj4R
22  end
```

The function `RjRdd` calculates the integrals $\langle R_{3d}(r)|j_k(qr)|R_{3d}(r)\rangle$ for $k = 0, 2, 4$. The input variable $q$ and the radial function $R(r)$ are in units of the Bohr radius ($a_0$). In lines 5-14, interpolating functions for $j_k(qr)$ are generated on a grid taken from $R$. The Bessel functions are calculated for $k = 0, 2,$ or 4. In lines 17-18, integrals $\langle R_{3d}(r)|j_k(qr)|R_{3d}(r)\rangle$ are calculated using interpolation functions. The result is returned on line 21.

We can now generate a function that creates the $\vec{q}$ dependent nIXS transition operator for $d - d$ transitions

```
23  -- potential expanded on spherical harmonics
24  function ExpandOnClm(k,theta,phi,scale)
25    ret={}
26    for m=-k, k, 1 do
27      table.insert(ret,{k,m,scale * SphericalHarmonicC(k,m,theta,phi)})
28    end
29    return ret
30  end
31  -- define nIXS transition operators
32  function TnIXS_dd(q, theta, phi, R)
33    Rj0R, Rj2R, Rj4R = RjRdd(q, R)
34    k=0
35    A0 = ExpandOnClm(k, theta, phi, I^k*(2*k+1)*Rj0R)
36    T0 = NewOperator("CF", NF, IndexUp_3d, IndexDn_3d, A0)
37    k=2
38    A2 = ExpandOnClm(k, theta, phi, I^k*(2*k+1)*Rj2R)
39    T2 = NewOperator("CF", NF, IndexUp_3d, IndexDn_3d, A2)
40    k=4
41    A4 = ExpandOnClm(k, theta, phi, I^k*(2*k+1)*Rj4R)
42    T4 = NewOperator("CF", NF, IndexUp_3d, IndexDn_3d, A4)
43    T = T0+T2+T4
44    T.Chop()
45    return T
46  end
```

The function `TnIXS_dd` generates and returns the nIXS transition operator, as a function of $q = (q, \theta_q, \phi_q)$ for a given radial function $R$. Similarly to the dipole transitions created for XAS in Sect. 5.3, the crystal field operator is used, generating the transition operator as a potential expanded on spherical harmonics.

Given the Hamiltonian and the eigenstates, calculated in Sec. 5.2, the nIXS spectra are calculated using `CreateSpectra`. In the previous example, the DFT radial functions are stored in the variable `radialFunctions`. This variable contains a list of radial functions as interpolation objects and the Ni $3d$ at the first position.

For example, the script creates the nIXS spectrum for $q = 4$ per $a^0$ in the $z$ direction.

```
47  T = TnIXS_dd(4, 0, 0, radialFunctions[1])
48  nIXSSpectra = CreateSpectra(Hamiltonian, T,
49  psiList, {{"Emin",-1}, {"Emax",6}, {"NE",3000}, {"Gamma",0.1}})
50  nIXSSpectra.Print({{"file","NiOnIXS_dd.dat"}})
```

## 5.6 Resonant Inelastic X-ray Scattering

Nonlinear response functions can be calculated using the function `CreateResonantSpectra`. For example, one can look at the $2p$ to $3d$ back to $2p$ resonant inelastic x-ray scattering (RIXS) in NiO, i.e. the $L_{2,3}M_{4,5}$ edge. To do this, we can start from the example script used for XAS in section 5.3. Calculating the RIXS spectra requires one additional line to the script.

```
69  RIXSSpectra = CreateResonantSpectra(Hamiltonian, Hamiltonian,
70        T2p3dx, T3d2py, psiList[1],
71        {{"Emin1",-10}, {"Emax1",20}, {"NE1",120}, {"Gamma1",1.0},
72         {"Emin2",-0.5}, {"Emax2",7.5}, {"NE2",8000}, {"Gamma2",0.05}})
```

### 5.7 Dynamical mean-field theory

QUANTY can be used to perform dynamical mean-field theory calculations. Green's functions can be calculated using the function `CreateSpectra`. The self-consistency loop can be done by Response function objects.

For a given energy grid `Grid = {E1,E2,E3,...}` and a single band Green function `G0` and local Coulomb interaction `U`, the following loop calculates the self-consistent dynamical mean field bath Green's function and self-energy.

```lua
for i=1,10 do
  -- Create Anderson impurity Hamiltonian on a grid
  GImpgrid = ResponseFunction.CalculateHybridizationFunction(G0,Sigma,
                                     {{"EnergyGrid",Grid}})
  -- transform the Response function to a natural impurity orbital
  -- representation
  GImpgridNat = ResponseFunction.ChangeType(GImpgrid,"Nat")
  -- create an operator from the response function
  HImp, BitField = ResponseFunction.ToOperator(GImpgridNat,
                                     {{"AddSpin",true}})
  -- Defien PES transition operator and local Hamiltonian
  TPESDn = NewOperator(HImp.NF,0,{{-0,1}})
  Hloc = NewOperator(HImp.NF,0,{{0,-0,1,-1,1},
                                {0,-0,-1/2},{1,-1,-1/2}})
  H = HImp + U * Hloc
  -- set restrictions
  StartRestrictions = {BitField.NF,BitField.NB,
    {BitField.ImpurityExtDn,(BitField.NBathExt+BitField.NImpurity)/4,
                            (BitField.NBathExt+BitField.NImpurity)/4},
    {BitField.ImpurityExtUp,(BitField.NBathExt+BitField.NImpurity)/4,
                            (BitField.NBathExt+BitField.NImpurity)/4},
    {BitField.BathVal     ,BitField.NBathVal   , BitField.NBathVal},
    {BitField.BathCon     ,0                    , 0}}
  GSRestrictions   = {BitField.NF,BitField.NB,
    {BitField.BathVal     ,BitField.NBathVal-1 ,BitField.NBathVal},
    {BitField.BathCon     ,0                   ,1}}
  PESRestrictions   = {BitField.NF,BitField.NB,
    {BitField.BathVal     ,BitField.NBathVal-2 ,BitField.NBathVal},
    {BitField.BathCon     ,0                   ,1}}
  -- Calculate ground-state
  psi = Eigensystem(H,StartRestrictions,1,
                    {{"Restrictions",GSRestrictions}})
  -- Calculate spectra
  GPESDnSpec, GPESDn = CreateSpectra(H, TPESDn,  psi,
          {{"Restrictions",PESRestrictions},{"NTri",NTri}})
  -- Sum PES and IPES to get G
  G = GPESDn[1] + ResponseFunction.InvertEnergy(GPESDn[1])
  -- Calculate Sigma
  Sigma = ResponseFunction.CalculateSelfEnergy(GImpgridNat,G)
end
```

Once a self-consistent self energy is calculated, the resulting Anderson impurity Hamiltonian can be used to calculate different spectral functions.

## 6 Graphical interface

The examples presented above highlight the extensive capabilities and the flexibility of QUANTY. However, the scripting language-based nature of the library can discourage potential users, even though, Lua has one of the most user-friendly syntax. In an effort to make QUANTY more accessible to the wider user community, a graphical user interface was devel-

oped independently at the European Synchrotron Radiation Facility (ESRF). The application, named CRISPY [98], is developed using Python and uses the Qt library. In addition, it relies on several open-source scientific libraries from the Python ecosystem, in particular the silx library, which provides the specialized Qt visualization widgets. Using CRISPY, the users can easily generate input files, submit calculations, and plot the resulting spectra for a wide-range of spectroscopies, edges, point group symmetries, and elements, using crystal field and ligand field Hamiltonians. The application can be installed on all major operating systems where Python is available, with prebuild and easy-to-use installers available for Windows and macOS. Additional documentation and precompiled installers are available on the official website.

## 7 Conclusion

Version 0.8 of QUANTY implements a set of functions that allows one to calculate a multiple of core level spectra for different levels of theory, namely crystal field, ligand field and dynamical mean field theory.

## Acknowledgements

We would like to thank all users of QUANTY who provided useful feedback on the code. The development of QUANTY has been funded by the DFG (German Research Foundation) – Project-ID 273811115 – SFB 1225 ISOQUANT and Project-ID 449872909 – FOR 5249 QUAST, by the EMPIR programme co-financed by the Participating States and from the European Union's Horizon 2020 research and innovation programme MetroMMC, "Measurement of fundamental nuclear decay data using metallic magnetic calorimeters". Part of the calculations are performed on the JUSTUS 2 cluster supported by the state of Baden Württemberg through bwHPC and the German Research Foundation (DFG) through grant no INST 40/575-1 FUGG (JUSTUS 2 cluster).

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
