# Peer review of "QUANTY, a quantum many-body scripting toolkit"

_SciPost Physics Codebases_

## Round 1 · Referee Report · Anonymous (Referee 1) · 2025-2-4

Strengths

  • The manuscript presents a mature codebase that is currently being used heavily in the community.
  • The paper is generally well written.
  • The code is accompanied by extensive online documentation.
  • Document provides a fairly comprehensive review of the theory needed to understand the code and its design philosophy.

Weaknesses

  • The paper has some issues with notation and clarity in selected parts.
  • The level of detail and background provided for some aspects is variable. A general user/reader would benefit from better referencing.

Report

The submitted manuscript is a code release for Quanty, a scripting toolkit for solving quantum many-body problems using exact diagonalization and Lanczos-based methods. The Quantity codebase is quite mature and well-known in the community, particularly as a platform for calculating core-level spectroscopies.

Quantity has a large user base and extensive documentation, which users can find on the projects' website. The submitted manuscript outlines the code's design philosophy and provides a few representative examples. Between the manuscript and existing online documentation, I believe the submission meets the submission requirements for SciPost Codebases, and I am inclined to recommend publication. However, I also have several comments that I would like the authors to address or consider, as given under "requested changes".

Requested changes

1) The presentation of the underlying theory provides some introductory material; however, I found the level of detail quite variable depending on the topic, and there were some notable omissions. For example, the paper never states the convention for normal ordering of the fermion operators in the section on second quantization and encoding the Fock states. Similarly, the DMFT example (Sec. 5.7) would benefit from a dedicated section explaining the self-consistency loop and specific equations being solved. From reading the document, It is also unclear whether Quanty only supports spin-1/2 fermions. To address the varying levels of detail, the authors might consider providing more references to relevant textbooks.

3) The document does not mention competing codes or discuss the relative strengths and weaknesses of those codes concerning Quanty. Some discussion on this point is warranted.

4) I suggest the authors remove bosons from the examples since they have not yet been fully implemented in the code. As it currently stands, bosons are discussed in some sections but not others, which leads to unnecessary confusion.

5) The authors could expand the conclusion section to include future development efforts. This section might be a good place to introduce the partially implemented boson support.

6) The text comments in the code snippits on page 14 are cut off. It would also be useful to explain whether or not Quanty expects this text to be included on the input line or whether it's a comment ignored by the software.

7) There are many places where the notation is mixed up or needs to be cleaned up. Here is a list of what I found.

Section 3.2: Please define H = H_1 + H_2 and distinguish H_1 here, which is the non-interacting Hamiltonian from H_1(t), which establishes the probe perturbation for the linear response calculations.

Eq. (18) is a second-order term, not a generic higher-order term, correct?

Section 3.3: I suggest changing "Measurements in physics and chemistry are based on ..." to "Measurements in physics and chemistry are often based on ...". Some would argue that RIXS is not a small perturbation.

Eq 20: I recommend using f and f' for the interior sums. The authors should also unify the notation here with the H_1 and H_2 notations highlighted in my previous comment.

Page 10 "Anser" should be "answer." I also recommend putting the "m" and "n" indicies in math mode.

Page 11: The elements of the tri-diagonal matrix are a_i while the operators are also denoted as a_i. The reader can figure out the meaning of each from context, but I recommend changing one to make things easier for the reader.

Page 25: Should the reference to line 20 really be to line 21?

Recommendation

Ask for minor revision

---

## Round 1 · Referee Report · Anonymous (Referee 2) · 2025-2-13

Strengths

  • The manuscript presents a matured codebase with a long history of use in the community.
  • The paper is generally well written.
  • The software is accompanied by extensive online documentation, detailing the use if used as a product.
  • The submitted introduces the physical background in an extensive manner.

Weaknesses

  • The notation can be improved, e.g. for using different fonts/typesetting for Lua objects, functions, etc. Some examples are given below requested changes.
  • Section 5.7: DMFT. It is remarkable, that a DMFT loop can be written in the internal scripting language. Nevertheless, I get the feeling, that this example could benefit from some more work. The code for the Start restrictions does not look very intuitively...
  • questionable non-standard build system. I can appreciate its simplicity, but also had my problems of compiling the software on two different linux distros.
  • Licensing the code under a CC-BY license, is generally not recommenden. Even Creative Commons themselves advise against using it for software: https://creativecommons.org/faq/#can-i-apply-a-creative-commons-license-to-software Therefore I recommend to choose a different license specifically for source code ( https://choosealicense.com/ ). I know, scipost allows CC-BY for software....
  • some guidance on the testsuite would be helpful also in the submitted document. I had to figure out the use of the tests from the tests that are executed in the CI pipeline.

Report

This submission concerns Quanty, a scripting language/framework for solving quantum many-body problems using ED and Lanczos methods.
Quanty already has a long history and is known as a platform for calculating core-level spectroscopies.

In its long history Quanty was able to grow its userbase using workshops, and an extensive online documentation of its scripting language has been created.

As far as this reviewer is concerned, the submission meets the submission requirements for SciPost Codebases. I could find tests for benchmarking and the documentation details multiple examples. The code follows its own set of programming standards to ensure quality, that I can at least partially understand.
The submitted manuscript outlines the code's design philosophy and details its use on various examples.
The submission includes instructions on downloading, installing and running the software.
Therefore I recommend publication in scipost codebases after the below changes have been addressed/improved.

Requested changes

I found some typos: - pg. 4: microsocpic - pg. 10: compete expected complete - pg. 19: multiple instances of Responsefunctions whereas on pg. 18 it's still response function. To indicate a data type, a different Font could be useful. A similar hting is on pg. 13, sec 4.1 with userdata , I believe. - pg. 26: "These are needed as an operator in second quantization is not defined for a specific configuration." What is meant here? - Some figures for the examples would enhance the representation. - Improve the contextualization of the software. So far there is no mention of similar projects, and where the strengths of Quanty wrt to the other codes.

Recommendation

Ask for minor revision

---

## Editorial Decision

awaiting_resubmission